# Improved Sampling Of Diffusion Models In Fluid Dynamics With Tweedie's Formula

**Youssef Shehata**[*]     **Benjamin Holzschuh**     **Nils Thuerey**
Technical University of Munich
85748 Garching, Germany

## Abstract

State-of-the-art Denoising Diffusion Probabilistic Models (DDPMs) rely on an expensive sampling process with a large Number of Function Evaluations (NFEs) to provide high-fidelity predictions. This computational bottleneck renders diffusion models less appealing as surrogates for the spatio-temporal prediction of physics-based problems with long rollout horizons. We propose *Truncated Sampling Models*, enabling single-step and few-step sampling with elevated fidelity by simple truncation of the diffusion process, reducing the gap between DDPMs and deterministic single-step approaches. We also introduce a novel approach, *Iterative Refinement*, to sample pre-trained DDPMs by reformulating the generative process as a refinement process with few sampling steps. Both proposed methods enable significant improvements in accuracy compared to DDPMs, DDIMs, and EDMs with NFEs $\leq 10$ on a diverse set of experiments, including incompressible and compressible turbulent flow and airfoil flow uncertainty simulations. Our proposed methods provide stable predictions for long rollout horizons in time-dependent problems and are able to learn all modes of the data distribution in steady-state problems with high uncertainty. [1]

## 1 Introduction

Turbulent flow is prevalent in everyday phenomena ranging from natural occurrences (Sullivan and McWilliams, 2024; Pyakurel et al., 2017) to engineering applications (Tulapurkara, 1997; Cheah et al., 2007; Volpe et al., 2014). Computational Fluid Dynamics (CFD) is essential for understanding these flows, with direct numerical simulations being the gold standard. However, it requires high-resolution grids to resolve the full spectrum of turbulent spatial and temporal scales, resulting in computationally intensive simulations (Pope, 2012). This limitation has propelled the recent surge in data-driven approaches. Leveraging the abundance of high- and low-fidelity data, machine learning algorithms offer various opportunities to enhance the accuracy and efficiency of turbulence simulations (Vinuesa and Brunton, 2022), particularly for phenomena like turbulent flows, which are challenging for traditional CFD methods.

**Diffusion models (DMs) as surrogates.** DMs (Hyvärinen, 2005; Sohl-Dickstein et al., 2015) have demonstrated great potential in various domains (Dhariwal and Nichol, 2021; Wang et al.; Lugmayr et al.; Ho et al.; Li et al., 2023); however, their application to fluid-based problems remains an underexplored area of research. To date, applications involved inverse problems (Holzschuh et al., 2023), high-fidelity reconstruction (Shu et al., 2023), autoregressive sampling in two-dimensional (Yang and Sommer, 2023; Lippe et al., 2023; Kohl et al., 2024) and three-dimensional (Lienen et al., 2024) settings, and an uncertainty-aware surrogate for airfoil simulations (Liu and Thuerey, 2024).

**Motivation for using DMs.** DMs have been shown to autoregressively generate videos or simulation trajectories, which are unconditionally stable over very long time horizons (Kohl et al., 2024). Temporal stability is difficult to accomplish using supervised loss terms, requiring memory-consuming techniques like multi-step unrolling (Um et al., 2020). Additionally, the probabilistic nature of DMs can deal very well with measurement noise or missing data (Huang et al., 2024), making them highly

---

[*]Correspondence to: `y.shehata@tum.de`
[1]The source code is available at `https://github.com/tum-pbs/tsm-ir-diffusion`.

robust and versatile. Since a prediction from the DM samples from the posterior, this allows small variations due to uncertainty in the input to naturally evolve and amplify over time, creating diverse predictions over many steps, while individual trajectories always remain physically accurate.

**Need for faster and more accurate sampling.** The main drawback of DMs, especially evident in fluid problems, is the long inference time due to the large Number of Function Evaluations (NFEs) required, and the limited accuracy compared to deterministic baselines (Cachay et al., 2023). Therefore, our objective in this study is to reduce the inference time disparity between DMs and single-step deterministic baselines while concurrently enhancing the accuracy of their predictions. This is achieved through our proposed straightforward training and sampling procedures.

The main contributions of this work can be summarized as follows:

1. We re-formulate the autoregressive problem to facilitate flexible sampling with extended parallelization and enable single networks to predict not only the next timestep but also intermediate states.
2. We introduce *Truncated Sampling Models (TSMs)* to enable single-step and few-step sampling while preserving or even augmenting sampling fidelity. We describe how truncation of the diffusion process, typically employed to reduce NFEs, can improve inference accuracy. Additionally, we distinguish our proposed TSMs from related approaches and highlight their efficiency and ease of implementation.
3. We introduce an inherently stochastic *Iterative Refinement (IR)* approach to enable flexible sampling of conditional diffusion models, allowing reduced NFEs with improved accuracy compared to ancestral sampling. We explain the intuition behind the approach and provide a comparative analysis against existing methods throughout our experiments.
4. We empirically demonstrate the efficacy of the proposed methods in reducing inference steps and improving the accuracy of diffusion models for fluid dynamics simulations through a diverse set of experiments, including compressible and incompressible turbulent flows in both time-dependent and steady-state settings.

## 2 BACKGROUND

### 2.1 DENOISING DIFFUSION PROBABILISTIC MODELS

Denoising Diffusion Probabilistic Models (DDPMs) (Ho et al., 2020), a class of generative DMs, convert a data distribution $q(x_0)$ to a prior distribution $q(x_T) \sim \mathcal{N}(0, \boldsymbol{I})$ over $T$ steps through a Markovian forward diffusion process $q(x_t \mid x_{t-1})$ by gradually adding Gaussian noise with noise schedule $\beta_t$. We can sample the state $x_t$ directly from $x_0$ through the parameterized closed form:

$$q(x_t \mid x_0) = \sqrt{\bar{\alpha}_t}x_0 + \sqrt{1 - \bar{\alpha}_t}\epsilon \, , \tag{1}$$

where $\bar{\alpha}_t = \prod_{i=1}^{t} \alpha_i$ and $\alpha_t = 1 - \beta_t$. The forward process posterior $q(x_{t-1} \mid x_t)$ is approximated using a neural network through a parameterized Gaussian distribution $p_\theta(x_{t-1} \mid x_t) = \mathcal{N}(x_{t-1}; \mu_\theta(x_t, t), \Sigma_\theta(x_t, t))$, where $\mu_\theta$ and $\Sigma_\theta$ are the network predicted mean and variance, respectively. However, in this study, the variance is kept constant according to the noise schedule $\beta_t$ and is chosen to be $\Sigma_\theta(x_t, t) = \sigma_t^2 \mathbf{I}$, with $\sigma_t^2 = \beta_t$.

During inference, the reverse process (i.e., ancestral sampling) begins from $x_T \sim \mathcal{N}(0, \boldsymbol{I})$ and iteratively samples $x_{t-1} \sim p_\theta(x_{t-1} \mid x_t)$ for $T$ steps until reaching a fully denoised state $x_0$. The network is trained to estimate the forward process posterior by minimizing the Kullback-Leibler (KL) divergence $\mathrm{KL}(q(x_{t-1} \mid x_t) \| p_\theta(x_{t-1} \mid x_t))$, which reduces to the loss function (Ho et al., 2020):

$$L_{t-1} := \mathbb{E}_{x_0, \epsilon} \left\| \epsilon - \epsilon_\theta(\sqrt{\bar{\alpha}_t}x_0 + \sqrt{1 - \bar{\alpha}_t}\epsilon, t) \right\|^2 , \tag{2}$$

where the network only learns to predict the noise at each noise step to perform partial denoising. The denoising step is related to Tweedie's formula (Efron, 2011), which can be used to estimate the posterior mean $\mathbb{E}[\hat{x}_0 | x_t; \theta]$ from a noisy sample $x_t$ via

$$\mathbb{E}[\hat{x}_0 | x_t; \theta] = (x_t - \sqrt{1 - \bar{\alpha}_t}\epsilon_\theta(x_t, t))/\sqrt{\bar{\alpha}_t}. \tag{3}$$

## 2.2 PROBLEM DEFINITION

The Navier-Stokes (NS) Partial Differential Equations (PDEs) represent a class of problems that epitomize the complex physics encountered in engineering and scientific disciplines. For an arbitrary domain $\Omega$, fluid motion is governed in space and time $\tau$ by the NS PDE, defined as:

$$\frac{\partial u}{\partial \tau} + (u \cdot \nabla)u = -\nabla p + \frac{1}{Re}\Delta u + f, \quad \frac{\partial \rho}{\partial \tau} + \rho(\nabla \cdot u) = 0, \tag{4}$$

where $f$ is external forcing, and $u$, $p$, and $\rho$ are the velocity, pressure, and density, respectively. $Re$ is the non-dimensional Reynolds number, controlling the severity of diffusive to convective transport.

**Reformulated autoregressive sampling.** For time-dependent problems, to reach the target state $\mathbf{x}(\tau_f)$, where $\mathbf{x} = \{u, p, \rho\}$, and with the initial condition $\mathbf{x}_0$, our reformulation of autoregressive sampling is defined as (notice that the notations $\mathbf{x}(\ldots)$ and $\mathbf{x}_{\ldots}$ are used interchangeably):

$$\mathbf{x}(\tau_f) = p_\theta^T(x_T, p_\theta^T(\ldots p_\theta^T(x_T, \mathbf{x}_0, j)\ldots), j), \tag{5}$$

where the prediction stride $j$ denotes how far we sample in the future based on the physical timestep $\delta\tau$. The shortened notation $p_\theta^T$ denotes the DDPM iterative sampling, $p_\theta^T(x_T, \mathbf{x}_0, j) = \mathbf{x}(j \cdot \delta\tau) = p_\theta(p_\theta(\ldots p_\theta(x_T, \mathbf{x}_0, j)\ldots), \mathbf{x}_0, j)$. In essence, $p_\theta^T$ maps any fluid state $\mathbf{x}(\tau)$ to $\mathbf{x}(\tau + j \cdot \delta\tau)$, without the need to estimate the intermediate states $\mathbf{x}(\tau + i \cdot \delta\tau) \,\forall i < j$, as typically required by classical numerical solvers to satisfy the Courant–Friedrichs–Lewy (CFL) convergence conditions (de Moura and Kubrusly, 2013). Although conditioning on $j$ has been previously explored in the context of multi-parameter conditioning (Gupta and Brandstetter, 2023), our contribution emphasizes the use of $j$ to facilitate flexible, parallelizable, and potentially more accurate diffusion sampling instead of next-step predictions.

**Flexibility in predicting future states.** By conditioning a surrogate model on $j$ for $j \in \{0, \ldots, \mathcal{T}\}$, we achieve two major benefits. First, the model can predict all possible intermediate states between $\tau = 0$ and $\tau = \mathcal{T} \cdot \delta\tau$. In comparison to methods such as *DYffusion* (Cachay et al., 2023) that require independent forecaster and temporal interpolator networks to achieve this task, our formulation enables a single network to predict the next timestep in addition to intermediate ones. Second, we are able to balance between the accuracy of the first-step prediction and error accumulation (Lienen et al., 2024). For instance, smaller $j$ values would lead to first-step predictions with high accuracy; however, they would require longer rollout steps, leading to more error accumulation and vice versa. Thus, we will demonstrate in our experiments how an optimal value for $j$ can lead to better accuracy than next-step sampling.

## 3 RELATED WORK

**DMs as flow surrogates** In inverse problems involving temporal evolution, Holzschuh et al. (2023) utilized DMs to predict a system's state backward in time, integrating an approximate inverse physics simulator into the sampling process. For super-resolution, Shu et al. (2023) applied DDPMs to turbulent flows by reconstructing high-fidelity flow fields from low-fidelity inputs, achieving remarkable results even with sparse input data. Furthermore, regarding time-dependent autoregressive predictions, Yang and Sommer (2023) attempted to predict nonlinear fluid fields at specific points in time based on initial conditions. Kohl et al. (2024) introduced an autoregressive conditional diffusion model (ACDM) capable of predicting fluid states over extended time horizons while maintaining sample quality and temporal stability, and provided benchmark results on various datasets and against multiple baselines. Lienen et al. (2024) explored spatio-temporal predictions in three-dimensional turbulent flows and achieved faster processing times than conventional solvers. Additionally, *PDE-refiner* facilitates the enhancement of sampling precision across all frequency components inherent in PDE solutions through a multistep refinement procedure (Lippe et al., 2023). Finally, an uncertainty-aware surrogate for steady-state airfoil turbulence was presented by Liu and Thuerey (2024) to predict the uncertainty of the simulations and provide samples from the learned ground truth distribution.

**Expedited sampling.** The slow sampling time of DMs is a major drawback, prompting extensive research to reduce the computational cost without compromising quality. Early endeavors included

learning the variances of the reverse process and optimizing the noise schedule (Nichol and Dhariwal, 2021). In tandem, Song et al. (2021) presented Denoising Diffusion Implicit Models (DDIMs) that generalize DDPMs via a class of non-Markovian diffusion processes instead of the Markovian diffusion process of DDPMs, leading to shorter deterministic generative processes. By expressing DMs in a common framework known as elucidated DMs (EDMs), Karras et al. (2022) introduce a design space featuring separable design choices that can be optimized to attain expedited sampling with state-of-the-art accuracy. Further, introducing Bespoke solvers represents a novel framework for crafting custom ordinary differential equation solvers tailored to pre-trained models, yielding parameter-efficient solvers with negligible training overhead (Shaul et al., 2024). Moreover, distillation techniques (Luhman and Luhman, 2021; Salimans and Ho, 2022; Meng et al., 2022; Sauer et al.) offer promising avenues for achieving high-fidelity image synthesis with diminished computation overhead through few steps and one-step inference, albeit at the cost of accuracy (Salimans and Ho, 2022; Meng et al., 2022). Analogously, Song et al. (2023) propose consistency models, trained via distillation or in isolation, to directly map noise to data and facilitate both single-step and few-step sampling. Finally, other endeavors include latent diffusion models (LDMs) (Rombach et al., 2022) and Truncated Diffusion Probabilistic Models (TDPMs) (Zheng et al., 2023). LDMs compress input states into latent spaces of reduced degrees of freedom through an additional encoder-decoder network, consequently reducing computational costs associated with DDPM sampling for the same/reduced NFEs. TDPMs truncate the last steps of the forward diffusion process, leading to a shorter generative process starting from a hidden noisy distribution by leveraging an additional generative adversarial network (GAN)-based implicit generator to match the prior to the aggregated posterior.

These diverse approaches collectively contribute to augmenting the generative process of DMs, addressing computational challenges, and enhancing sample quality. The two approaches that we introduce offer complementary solutions, particularly in the domain of physics-based simulations, and provide insights for the development of DMs in general.

## 4 NOVEL TRAINING AND SAMPLING APPROACHES

### 4.1 TSM: TRUNCATED SAMPLING MODEL

**Motivation.** An interesting phenomenon of DDPMs, particularly when trained with a linear $\beta_t$, is the ability to skip a small percentage (i.e., $\leq 20\%$) of the reverse diffusion process while maintaining the sampling quality (Nichol and Dhariwal, 2021). Furthermore, approaches have been devised to target a relevant range of noise levels during training by prioritizing intermediate noise levels (Karras et al., 2022; Choi et al., 2022), thereby enhancing the loss per noise level.

We re-visit these approaches with a new perspective: We truncate a significant part from the last steps of the reverse Markov chain with a high skip percentage $s$ to reduce NFEs and focus the training on noise steps preceding the truncation. We refer to a model trained for a limited part of the diffusion process and sampled with truncated ancestral sampling as *Truncated Sampling Model (TSM)*. Focused training by restricting the sampling window for $t$ parallels approaches by Karras et al. (2022) and Choi et al. (2022) to improve the loss per noise level (see Eq. 2), with our main objective to achieve enhanced sampling accuracy. Hence, we expect $s \gg 0$ to lead to better accuracy and reduced NFEs.

**Sampling.** Algorithm 1 summarizes the sampling procedure for conditional TSMs, with differences from ancestral sampling highlighted in blue. TSM sampling follows ancestral sampling until $x_{t_s}$, i.e., $p_\theta(x_{t_s:T}) := p(x_T) \prod_{t=t_s+1}^{T} p_\theta(x_{t-1}|x_t)$, where $t_s = \lfloor s \cdot T \rfloor$ and $s \in (0,1]$. At $t = t_s$, instead of sampling $x_{t_s-1} \sim p_\theta(x_{t_s-1}|x_{t_s})$, we estimate $\hat{x}_0$ using the posterior mean $\mathbb{E}[\hat{x}_0|x_t; \theta]$, see Eq. 3.

**Training.** The TSM training procedure involves the choice of the hyperparameter $s$ (skip percentage) and its use as a lower bound for sampling diffusion steps. Hence, the sole adjustment to the DDPM training algorithm from Ho et al. (2020) is defined as $t \sim \text{Uniform}(\{t_s, \ldots, T\})$. Based on the skip percentage $s$, typically $> 0.2$ for more pronounced outcomes, a balance between NFEs, sampling accuracy, and stochasticity can be achieved. Extreme skip percentages (e.g., $s \approx 1$) are feasible, resulting in unprecedented high-accuracy single-step diffusion sampling, albeit at the cost of reduced stochasticity. Hence, relatively lower $s$ values are optimal for problems of stochastic nature to enable learning all modes of the data distribution.

---

**Algorithm 1** Truncated Ancestral Sampling for conditional TSMs

---

**Require:** $\epsilon_\theta$ (TSM), $s$ (skip percentage), $c$ (condition)
1: $x_T \sim \mathcal{N}(0, \mathbf{I})$
2: **for** $t = T, ..., t_s + 1$ **do**
3:      $z \sim \mathcal{N}(0, \mathbf{I})$ if $t > 1$, else $z = 0$
4:      $x_{t-1} = \frac{1}{\sqrt{\alpha_t}} \left( x_t - \frac{\beta_t}{\sqrt{1-\bar{\alpha}_t}} \epsilon_\theta(x_t, t, c) \right) + \sigma_t z$          ▷ DDPM sampling $p_\theta(x_{t-1} \mid x_t)$.
5: **end for**
6: **if** $s > 0$ **then** $\hat{x}_0 = (x_{t_s} - \sqrt{1 - \bar{\alpha}_{t_s}} \epsilon_\theta(x_{t_s}, t_s, c)) / \sqrt{\bar{\alpha}_{t_s}}$
7:                                ▷ Tweedie's formula $\mathbb{E}[\hat{x}_0 | x_t; \theta]$ (see Eq. 3).
8: **return** $\hat{x}_0$

---

**TSMs vs TDPMs.** We highlight the main differences between TDPMs (Zheng et al., 2023) and our proposed TSMs. First, TDPMs apply the truncation to the last steps of the diffusion process, whereas our approach focuses on these last steps and conversely truncates the first steps. Second, TDPMs rely on an additional implicit generator network (GAN) to match the prior with the aggregated posterior. This implicit generator requires joint training with the DDPM, thus adding extra complexity to the training process. On the other hand, our training procedure closely resembles the straightforward DDPM training procedure and doesn't add complexity, thus rendering TSMs an appealing approach for enhancing sampling efficiency. Finally, while the sample quality of TDPMs is adversely impacted by large truncations, TSMs enable one-step inference with the same or improved sampling quality, relying on characteristics specific to fluid dynamics datasets as we explain in Section 5.

### 4.2 IR: ITERATIVE REFINEMENT

**Motivation.** A fundamental characteristic of DMs is their iterative sampling process, which systematically reduces noise from a pure Gaussian noise $x_T$ to a noise-free state $x_0$ over successive iterations. This ancestral sampling procedure is performed by gradually estimating all intermediate states $x_{T-1:1}$ until reaching a fully denoised state $x_0$, matching the training data distribution $q(x_0)$. Nevertheless, during training, a DDPM network learns to only predict the noise field at each noise level of a predetermined $\beta_t$ independent of any adjacent states (see Eq. 2). Therefore, for a pre-trained DDPM model, various sampling methods can be employed without re-training, as long as they pertain to $\beta_t$ originally used for training.

We leverage this property of DMs to introduce an intuitive refinement algorithm as a novel sampling method for DDPMs, which we refer to as *Iterative Refinement (IR)*. In IR sampling, we consider a much shorter noise schedule $\gamma = \{t_r, \ldots, t_e\} \subset \{T, \ldots, 1\}$ and interpret the different noise levels to essentially correspond to different levels of detail for a given state. We believe that, for any provided initial state $x_{\text{init}}$, there exists a sequence $\gamma$ that defines the minimal number of noise levels (or levels of detail) sufficient to augment the accuracy of $x_{\text{init}}$. Hence, we optimize $\gamma$ to ensure that the accuracy of the final prediction closely matches all levels of detail present in the ground truth state $x_0$.

**Sampling.** Algorithm 2 summarizes the generative process for IR. Given an initial state $x_{\text{init}}$ (assumed to be a noise-free, low-order approximation of $x_0$) and a refinement schedule $\gamma$, we iteratively apply forward diffusion followed by an estimation of the posterior mean to predict a series of gradually enhanced approximations of the noise-free state $\hat{x}_0^i, \forall i = 1, \ldots, N$, where $N = |\gamma|$, at distinct noise levels. Conversely, ancestral sampling $p_\theta(x_{t-1}|x_t)$ gradually removes noise for the same schedule (assuming $\gamma = \{T, \ldots, 1\}$) leading to intermediate states that are partially noisy. IR allows for a flexible choice of the length and distribution of

---

**Algorithm 2** IR sampling procedure

---

**Require:** $\epsilon_\theta$ (DDPM model), $x_{\text{init}}$ (initial state), $\gamma = \{t_r, \ldots, t_e\}$ (refinement schedule)
1: $\hat{x}_0 \leftarrow x_{\text{init}}$
2: **for** Each $t$ in $\gamma$ **do**
3:      $\epsilon \sim \mathcal{N}(0, \mathbf{I})$
4:      $x_t = \sqrt{\bar{\alpha}_t} \hat{x}_0 + \sqrt{1 - \bar{\alpha}_t} \epsilon$
5:      ▷ Forward diffusion $q(x_t|x_0)$ using Eq. 1.
6:      $\hat{x}_0 = \frac{1}{\sqrt{\bar{\alpha}_t}} \left( x_t - \sqrt{1 - \bar{\alpha}_t} \epsilon_\theta(x_t, t, c) \right)$
7:      ▷ Tweedie's formula $\mathbb{E}[\hat{x}_0|x_t; \theta]$ from Eq. 3.
8: **end for**
9: return $\hat{x}_0$

---

$\gamma$ to fit the problem under consideration and ensures higher accuracy of predictions using fewer NFEs compared to ancestral sampling. We also argue that $x_{\text{init}}$ can be a (partially) noisy state obtained

through truncation of ancestral sampling, for example, or even sampled from the prior distribution $q(x_T)$, further relaxing the computational overhead before IR sampling.

**Optimizing the refinement schedule $\gamma$.** The most critical component of IR sampling is $\gamma$, balancing between inference quality and NFEs. Hence, an optimized $\gamma$ should consider the initial state $x_{\text{init}}$, the model's accuracy $L_{t-1}$ at each noise step (defined as in Eq. 2), and the nature of the problem. All our proposed $\gamma$ schedules are optimized using a greedy algorithm, which we found to easily lead to highly satisfactory results with minimal effort and computational cost. More details regarding $\gamma$ optimization are presented in Appendix B. A direct consequence of using this greeding algorithm is that the final output $\hat{x}_0^N$ provides a better approximation of $x_0$ than $x_{\text{init}}$ and all preceding approximations:

$$\mathbb{E}[\|\hat{x}_0^N - x_0\|_2] < \mathbb{E}[\|\hat{x}_0^{N-1} - x_0\|_2] < \ldots < \mathbb{E}[\|\hat{x}_0^1 - x_0\|_2] < \mathbb{E}[\|x_{\text{init}} - x_0\|_2]. \quad (6)$$

We believe that better results could be achieved through more sophisticated optimization of $\gamma$, although this will incur additional training overhead.

**Method novelty.** While IR makes use of Tweedie's formula to obtain an intermediate approximation of the posterior mean $\hat{x}_0$, it differs from the various resampling methods for posterior sampling (Song et al., 2024; Zhang et al., 2024) in the following aspects. (1) Freedom to choose $x_{init}$ and optimize a refinement schedule $\gamma$. While all sampling methods begin the inference procedure by sampling $x_T \sim \mathcal{N}(0, \mathbf{I})$, IR is more flexible as it enables the starting point $x_{init}$ to be $x_T$, a low-fidelity prediction, or a partially noisy output. (2) IR efficiently samples fluid states using a pre-trained model without the need for an auxiliary neural network or an optimization task to enforce data consistency to $\hat{x}_0$. This reduces the computational cost during sampling and introduces flexibility in choosing $\gamma$ without restrictions on requiring the estimates of $\hat{x}_0$ to be evaluated at noise steps close to 0 to ensure a meaningful prediction (Yu et al., 2023). Further, while PDE-Refiner (Lippe et al., 2023) shares similarities with DDPMs and IR, our method differs in key aspects. Unlike training from scratch with a fixed schedule, IR is a sampling algorithm for pre-trained DDPMs. This enables flexibility in choosing an optimal combination of $\gamma$ and $x_{init}$, making it both a standalone sampling algorithm (when $x_{init} \sim \mathcal{N}(0, I)$) and a refinement strategy for noisy or low-fidelity inputs.

**Relation to DDIMs.** The general recursive sampling formula is defined by Song et al. (2021):

$$x_{t-1} = \sqrt{\bar{\alpha}_{t-1}} \underbrace{\left( \frac{x_t - \sqrt{1 - \bar{\alpha}_t} \epsilon_\theta^{(t)}(x_t)}{\sqrt{\bar{\alpha}_t}} \right)}_{\text{predicting } x_0} + \underbrace{\sqrt{1 - \bar{\alpha}_{t-1} - \sigma_t^2} \cdot \epsilon_\theta^{(t)}(x_t)}_{\text{direction pointing to } x_t} + \sigma_t \epsilon_t, \quad (7)$$

with $\sigma_t = 0$ resulting in a deterministic forward process. One can see that Eq. 7 consists of two main components, a prediction of $x_0$ and a direction pointing to $x_t$, exactly matching IR sampling from Algorithm 2 with a reversed order. For deterministic DDIMs, the predicted noise $\epsilon_\theta$ in Eq. 7 is used to define the forward process instead of a randomly sampled Gaussian noise. This directly results from their choice of the mean function for the inference distribution $q(x_{t-1} \mid x_t, x_0)$ (refer to Song et al. (2021) for more details). Despite the similarity, our results demonstrate that IR consistently outperforms deterministic DDIM sampling. We hypothesize that the stochastic nature of IR aids in rectifying errors incurred in earlier sampling steps. However, the dynamics of stochastic sampling are complicated in practice and might introduce additional errors (Karras et al., 2022). Consequently, this observation may not generalize to other datasets and domains.

## 5 EXPERIMENTS

### 5.1 TEST CASES

We consider two-dimensional (2D) fluid flow test scenarios, including compressible transonic flow (`Tra`), incompressible forced turbulence (`Fturb`), and steady-state airfoil turbulence uncertainty (`Air`), as shown in Fig. 1. Details regarding all datasets can be found in Appendix A. These cases were selected to ensure diversity (similar to *PDEBench* (Takamoto et al., 2022)) and facilitate the assessment of various method aspects, such as temporal stability and stochasiticity. For the transient (i.e. time-dependent) cases, the prediction stride is defined as $j \in \{0, 1, \ldots, 10\}$, i.e., $\mathcal{T} = 10$, and is provided as an additional input channel to the network. Details regarding training and diffusion-related hyperparameters for all test cases can be found in Appendix C.

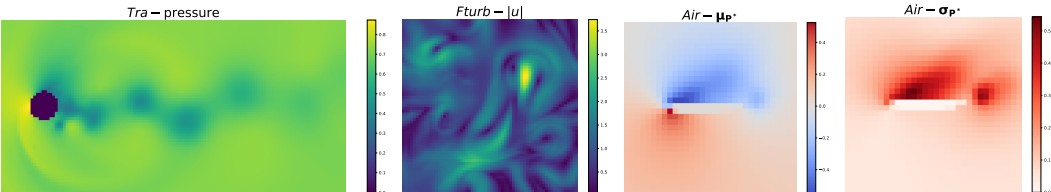

Figure 1: Showcasing diverse flow fields from our test cases. $p^*$ is the dimensionless pressure.

**Transonic flow (`Tra`).** Benchmark dataset by Kohl et al. (2024) for compressible transonic flow over a cylinder on a $128 \times 64$ grid including $u_x$, $u_y$, $p$, and $\rho$ flow fields. The case becomes particularly challenging at high Mach Numbers $Ma$ due to the presence of shock waves. Interpolation `int` and extrapolation `ext` datasets for evaluation involve $R = 60$ timesteps with $Ma \in \{0.66, 0.67, 0.68\}$ and $Ma \in \{0.50, 0.51, 0.52\}$, respectively. Temporal stability is tested on `Tra`$_{\text{long}}$ with $R = 240$.

**Forced turbulence (`Fturb`).** We generate the dataset by solving the incompressible NS PDE with a sinusoidal forcing term to obtain forced turbulence, i.e., Kolmogorov flow. We consider a range of $Re$ values with $Re = \{200, 1000, 2500, 4000\}$ for training, while testing is split into interpolation (`int`: $Re = \{1750\}$) and extrapolation (`ext`: $Re = \{100, 5000\}$) regions. The dataset includes $u_x$ and $u_y$ flow fields with $64 \times 64$ resolution. Autoregressive sampling is employed for $R = 30$ timesteps for two sequences per $Re$. An additional dataset `Fturb`$_{\text{long}}$ with $R = 120$ is considered to assess temporal stability of the models (see Appendix A for more details). We train a deterministic single-step baseline for comparison, using the same architecture as DDPMs without conditioning on $x_t$ or $t$, while using identical training parameters (see Appendix C).

**Airfoil turbulence uncertainty (`Air`).** The benchmark dataset by Liu and Thuerey (2024) models the inherent uncertainty of steady-state airfoil flow simulations with various airfoil profiles, $Re$ values, and angles of attack. For each combination of parameters, 25 solutions are obtained through different solver settings (e.g., number of iterations), representing the data uncertainty. Two main studies are considered. (1) `Air`$_{\text{One}}$ considers the uncertainty due to $Re$ only while keeping other parameters fixed (2) `Air`$_{\text{Multi}}$ considers the uncertainty arising from all parameters. `Air`$_{\text{Multi}}$ is limited to 1250 different configurations and both studies are restricted to a $32 \times 32$ resolution.

## 5.2 RESULTS

**Evaluation metrics.** We evaluate the accuracy of DMs against the ground truth data through various metrics. For the `Fturb` and `Tra` cases, we consider the temporal-average MSE (TA-MSE), turbulent kinetic energy spectrum (TKE), domain-wide kinetic energy (DWKE), temporal correlation $\rho$ for the absolute velocity $|u|$, and temporal stability for significantly long rollout horizons. For results obtained with $j > 1$, we restrict the analysis in this section to sampling with the stride $j$ without sampling the intermediate states (i.e., we only make $\lceil R/j \rceil$ predictions to reach the target timestep). Details regarding the sampling of the intermediate states are presented in Appendix D. In the `Air` case, the focus is on evaluating how accurately the data distribution, characterized by the mean $\boldsymbol{\mu}$ and standard deviation $\boldsymbol{\sigma}$ for each channel $\boldsymbol{y}$, is captured by the surrogates by evaluating $(\text{MSE}_{\boldsymbol{\mu}_y})_a$ and $(\text{MSE}_{\boldsymbol{\sigma}_y})_a$, where $(\cdot)_a$ denotes the average of a field.

We focus our evaluation on the speedup obtained via a reduction in NFEs, relying on the fact that we utilize the same architectures with almost the same number of parameters for all models. Training and sampling for all test cases were carried out using NVIDIA GeForce RTX 2080 Ti GPU. This section is solely dedicated to quantitative analysis of the top-performing models. We provide our comprehensive set of results as well as other baselines (such as ResNet, Fourier neural operators (FNO), PDE-Refiner, latent-space transformers (TF), and Heteroscedastic models) in Appendix E.1. Qualitative samples are provided in Appendix E.2. Moreover, we compare our methods against DDPMs, DDIMs, and EDMs and exclude other expedited sampling methods outlined in Section 3 based on the following considerations. Distillation techniques require extensive retraining without yielding significant accuracy improvements over the teacher model. Bespoke solvers introduce excessive complexity and resource demands to manage varying strides $j$ in our autoregressive formulation.

**Top performing models for transient cases.** We present TA-MSE results for the `Tra` and `Fturb` cases in Table 1. Using $j = 1$, our approaches significantly surpass DDPMs and DDIMs accuracy while requiring only a fraction of the NFEs, as low as single-step inference, whereas EDM exhibits variable performance. In `Tra` (see Table 1, left), our approaches transcend the benchmark ACDM (Kohl et al., 2024) with $\times 4$ and $\times 20$ (i.e., single-step inference) speedup for IR and TSM, respectively. Additionally, the single-step TSM outperforms the best baseline from Kohl et al. (2024), while IR yields marginally lower accuracy. The most accurate EDM is reported here, with comprehensive tests for various samplers detailed in Fig. 7 (Appendix E.1), demonstrating marginal increase in accuracy compared to TSM, albeit requiring $4\times$ the NFEs.

Regarding the `Fturb` case (see right of Table 1), while single-step TSMs demonstrate sufficient accuracy compared to expensive DDPMs and DDIMs, they prove suboptimal to the baseline. Although the model "TSM T80 $s = 0.75$" with 20 NFEs (see Table 8 in Appendix E.1) marginally outperforms the baseline, its increased computational cost renders it less favorable for our objective. In contrast to `Tra`, IR sampling with 10 NFEs provides the most accurate results, superseding the baseline, 80-step DDPM, 40-step DDIM, 10-step EDM, and 20-step TSM. Also, EDM is found to be on par with the single-step TSM in terms of accuracy but is ten times slower.

**Optimal $j$.** For optimal $j$ values greater than 1 in both test cases (see Appendix E.1), all models, except EDMs, reduce the TA-MSE significantly compared to values obtained by $j = 1$. While this improvement is seen only when sampling with the main stride $j$, we demonstrate in Appendix D how sampling the intermediate states could further reduce the TA-MSE, especially for large $j$.

**Temporal stability and physics-based metrics.** In addition to TA-MSE, we examine the correlation between the predicted $|u|$ to the ground truth over time for both `Tra` and `Fturb` cases. As shown in Fig. 8, top TSM and IR models with the lowest TA-MSE consistently exhibit high correlation to the ground truth across all timesteps, further confirming the superiority of our approaches in improving over DDPMs. Also, in Fig. 9, we showcase the temporal stability of the models over extended rollouts as they maintain physically-consistent results even after the predictions have decorrelated from the ground truth. Furthermore, as shown in Fig. 2, the TKE and DWKE plots indicate that both physics-based metrics are consistent with the TA-MSE results presented in Table 1, demonstrating agreement in models accuracy, except for the top performance by EDM for the DWKE metric.

**Foundations of our method's performance.** Our methods deliver superior results due to the effective use of Tweedie's formula in conjunction with the characteristics of the data distribution. While Tweedie's formula introduces biases for multimodal distributions, we demonstrated their applicability to predominantly unimodal distributions in fluid dynamics (Shu et al., 2023; Yang and Sommer, 2023). As a result, its application in our work does not lead to any bias, leading to single- and few-step sampling with the same or improved accuracy. Moreover, since typical fluid datasets are limited to relatively coarse resolutions, they inherently lack high-frequency details, allowing the truncation of the last steps in the reverse process without loss of information, see Fig. 2a.

**`Air` top performing models.** Table 2 summarizes the top results for $(\mathrm{MSE}_{\boldsymbol{\mu}_y})_a$ and $(\mathrm{MSE}_{\boldsymbol{\sigma}_y})_a$ in both `One` and `Multi` cases. In the `One` case, our enhancement over the benchmark from Liu and Thuerey (2024) is achieved by using a DDPM with a linear $\beta_t$ and normalizing the dataset.

Table 1: TA-MSE values for the top performing models in the `Tra` (left) and `Fturb` (right) cases. The table shows how our TSMs and IR sampling can yield highly accurate results with NFEs $\leq 5$ for `Tra` and NFEs $\leq 10$ for `Fturb`. UNet$_{ut}$ refers to UNet with unrolled training. In IR, T100/T80 is the base model, $\mathcal{N}$ is Gaussian noise used as $x_{\mathrm{init}}$, and the absence of $\gamma$ implies linear sampling steps. Standard deviation values are estimated over all timesteps and multiple samples.

| | | Tra $(10^{-3})$ | | | | Fturb $(10^{-2})$ | |
| --- | --- | --- | --- | --- | --- | --- | --- |
| Model | NFEs | ext | int | Model | NFEs | ext | int |
| ACDM T20 (Kohl et al., 2024) | 20 | $2.3 \pm 1.4$ | $2.7 \pm 2.1$ | | | | |
| UNet$_{ut}$ (Kohl et al., 2024) | 1 | $1.6 \pm 0.7$ | $1.5 \pm 1.5$ | Baseline | 1 | $3.95 \pm 3.84$ | $4.82 \pm 5.23$ |
| DDPM T100 | 100 | $3.0 \pm 2.7$ | $4.1 \pm 3.7$ | DDPM T80 | 80 | $6.16 \pm 6.54$ | $5.27 \pm 5.55$ |
| EDM - Deterministic Euler | 4 | $\mathbf{1.3 \pm 1.3}$ | $\mathbf{1.1 \pm 1.0}$ | EDM - Stochastic Euler | 10 | $4.75 \pm 4.53$ | $4.78 \pm 4.26$ |
| DDIM T20 | 10 | $3.2 \pm 2.7$ | $4.2 \pm 3.9$ | DDIM T80 | 40 | $4.31 \pm 4.62$ | $6.97 \pm 6.84$ |
| IR T100 - $\mathcal{N}$ $\gamma_1$ (Ours) | 5 | $1.6 \pm 1.3$ | $2.0 \pm 1.7$ | IR T80 - $\mathcal{N}$ (Ours) | 10 | $\mathbf{2.93 \pm 3.34}$ | $\mathbf{1.70 \pm 1.63}$ |
| TSM T100 $s = 1$ (Ours) | 1 | $\mathbf{1.2 \pm 1.1}$ | $1.5 \pm 1.5$ | TSM T100 $s = 1$ (Ours) | 1 | $5.00 \pm 5.52$ | $4.39 \pm 4.76$ |

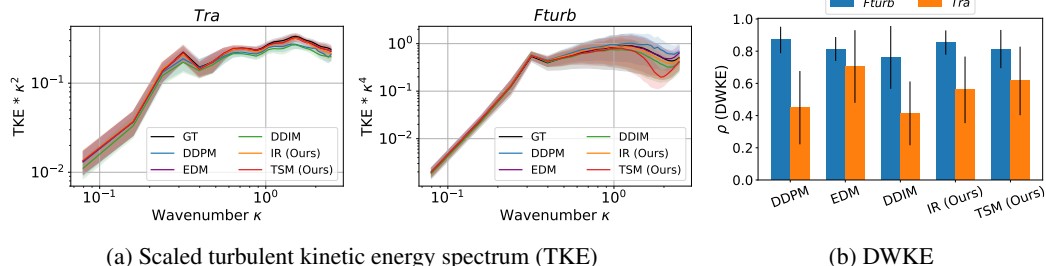

(a) Scaled turbulent kinetic energy spectrum (TKE)

(b) DWKE

Figure 2: **(a)** TKE = $E(\kappa) = |\tilde{u}(\kappa)|^2/N$, where $N$ is the total number of nodes in the domain and $\tilde{u} = \mathcal{F}(u)$, i.e., the Fourier transform of the velocity $u$. **(b)** Correlation coefficient $\rho$ for the time evolution of the DWKE, which describes the total kinetic energy of the system as DWKE$(u) = \frac{1}{N}\sum_{i=1}^{N}|u_i|^2/2$. High $\rho$ means the predicted states highly-correlate with the ground truth with respect to the temporal evolution of the DWKE. Both plots are based on the entire test dataset.

Table 2: `Air` test case results for the top-performing models. Our proposed models (TSM and IR) for `Air`$_{\text{One}}$ (left) provide highly accurate estimation of the data distribution for both $\boldsymbol{\mu}$ and $\boldsymbol{\sigma}$ with up to $\times 20$ speedup. In `Air`$_{\text{Multi}}$ (right), only the TSM supersedes the benchmark model in both parameters. In IR, $s = 0.6$ refers to $x_{\text{init}}$ obtained by truncated sampling. *Model uses cosine $\beta_t$.

| `Air`$_{\text{One}}$ $(10^{-4})$ | | | | `Air`$_{\text{Multi}}$ $(10^{-3})$ | | | |
|---|---|---|---|---|---|---|---|
| Model | NFEs | $(\text{MSE}_{\boldsymbol{\mu}_y})_a$ | $(\text{MSE}_{\boldsymbol{\sigma}_y})_a$ | Model | NFEs | $(\text{MSE}_{\boldsymbol{\mu}_y})_a$ | $(\text{MSE}_{\boldsymbol{\sigma}_y})_a$ |
| DDPM T200C* (Liu and Thuerey, 2024) | 200 | $3.79 \pm 0.27$ | $8.40 \pm 0.69$ | DDPM T200C* (Liu and Thuerey, 2024) | 200 | $1.76 \pm 0.23$ | $0.89 \pm 0.12$ |
| DDPM T200 | 200 | $2.88 \pm 0.26$ | $7.05 \pm 0.22$ | DDPM T100 | 100 | $2.34 \pm 1.11$ | $\mathbf{0.57 \pm 0.03}$ |
| EDM - Deterministic Heun | 20 | $8.13 \pm 1.28$ | $10.1 \pm 1.67$ | EDM - Stochastic Euler | 40 | $\mathbf{1.13 \pm 0.39}$ | $0.67 \pm 0.11$ |
| DDIM T200 | 100 | $3.68 \pm 0.44$ | $7.24 \pm 0.25$ | DDIM T100 | 50 | $2.12 \pm 0.99$ | $0.64 \pm 0.05$ |
| IR T100 - $s = 0.6$ $\gamma_5$ (Ours) | 41 | $\mathbf{2.87 \pm 0.32}$ | $6.76 \pm 0.20$ | IR T100 - $s = 0.6$ $\gamma_5$ (Ours) | 41 | $2.07 \pm 0.99$ | $0.59 \pm 0.04$ |
| TSM T100 $s = 0.9$ (Ours) | 10 | $3.30 \pm 0.39$ | $\mathbf{5.89 \pm 0.33}$ | TSM T100 $s = 0.75$ (Ours) | 25 | $1.66 \pm 0.53$ | $0.64 \pm 0.04$ |

Comparable accuracy is obtained using 10 NFEs (i.e., $\times 20$ speedup) by our TSM, whereas the best results are achieved by IR with 41 NFEs followed by suboptimal results from DDIM with 100 NFEs. Furthermore, Fig. 4a depicts the data distribution over increasing $Re$, backing up our claims regarding the accuracy of the low-NFEs models. EDMs perform poorly on this dataset, and exhibit difficulties converging to smooth target states. However, for `Multi`, they achieve the lowest errors for $\mu_y$ and are only competitive for $\sigma_y$ with NFEs $\geq 40$, see Fig. 10 (right). Notably, TSM requires at least 25 NFEs ($\times 2.5$ more than in `One`) to learn the full data distribution with high accuracy, while DDPM and DDIM models yield optimal results with ½ the NFEs used in `One`. We believe further hyperparameter tuning could potentially improve the TSM results with less NFEs.

**Further analysis of TSMs.** We analyze the effect of $s$ on the accuracy of TSMs compared to ancestral sampling without truncation for all test cases. As shown in Fig. 3, TSMs unconditionally outperform or at least match the accuracy of DDPMs with much fewer NFEs, depending on $s$. Substantial enhancement in accuracy with few-step sampling (i.e., high $s$) is made possible in the `Tra` and `Fturb` cases; however, in the `Air` test case, for the top performing linear $\beta_t$-based models with $s > 0.5$, we are only able to reduce NFEs for the same accuracy of DDPM. Noteworthy, for $s = 1$ in the `Air` case, the output is slightly noisy regardless of the satisfactory accuracy. For the cosine $\beta_t$-based model, we evidently see significant improvement to DDPM, which performs quite poorly and, accordingly, even the best TSM results for this case exhibit inadequate accuracy.

**Further analysis of IR.** In Fig. 4b, we compare IR against DDIMs with varying NFEs for identical sampling schedules in transient cases. Besides IR's consistency in transcending DDIMs across various base DDPMs, we notice a discernible correlation between NFEs and accuracy for DDIMs, whereas IR does not manifest a noticeable trend. The stochastic nature of IR makes it difficult to find clear relations across different datasets and NFEs as it is case-by-case tuned (Karras et al., 2022).

**Limitations.** In TSMs, since the training was focused on a limited part of the diffusion process, they exhibit inflexible sampling; neither DDIMs nor IR could be applied unless the sampling steps $t = \{t_{\text{start}}, \dots, t_{\text{end}}\} \subset \{T, \dots, t_s\}$, which is not the case for high $s$ values. Also, while IR and TSMs enable single- and few-step sampling with increased accuracy compared to ancestral sampling,

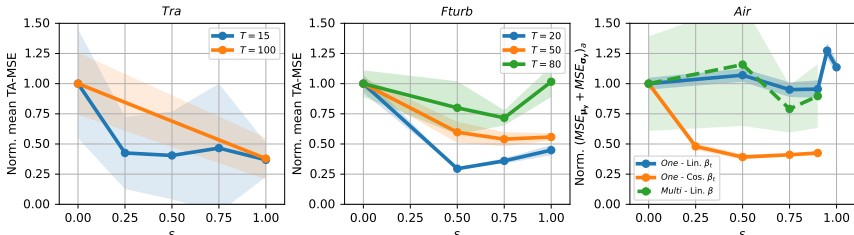

Figure 3: Effect of $s$ on TSMs performance for all test cases using $j = 1$ (for transient problems). Evaluation metrics are normalized by the $s = 0$ value; hence, values lower than 1 demonstrate higher accuracy of TSM compared to DDPM with no truncation. The shaded regions represent the standard deviation from both `ext` and `int` regions, and multiple samples.

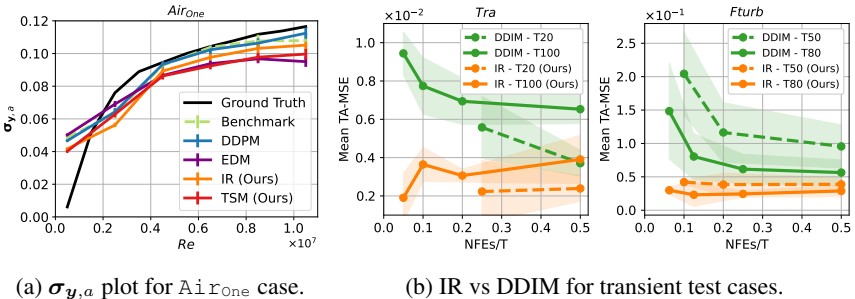

(a) $\boldsymbol{\sigma}_{\boldsymbol{y},a}$ plot for `Air`$_{\text{One}}$ case.     (b) IR vs DDIM for transient test cases.

Figure 4: **(a)** Comparing the average $\boldsymbol{\sigma}_{\boldsymbol{y}}$ of the models presented in Table 2 for the `Air`$_{\text{One}}$ case with increasing *Re*. Ground truth and Benchmark plots are obtained from Liu and Thuerey (2024). **(b)** Comparing the sampling accuracy of IR against DDIM for the `Tra` (left) and `Fturb` (right) cases using $j = 1$ and the same linear sampling schedule. The plot shows consistency by IR to provide better estimations than DDIM for both base DDPMs. The shaded regions represent the standard deviation from both `ext` and `int` regions, and multiple samples.

they are not guaranteed to supersede the accuracy of deterministic baselines. However, we have empirically shown that our methods improve over DDPMs, DDIMs, and EDMs in terms of speed and/or accuracy across our diverse datasets.

## 6 CONCLUSION

We have introduced two novel training and sampling approaches to enable single- and few-step sampling of DDPMs without compromising inference quality. Our first contribution is a Truncated Sampling Model (TSM), capable of achieving single-step inference while maintaining or even enhancing accuracy through early truncation of the reverse process. Additionally, our second contribution, Iterative Refinement (IR), targets pre-trained DDPMs by formulating the sampling process as a refinement endeavor to facilitate high-fidelity inference with reduced NFEs compared to existing sampling methods, such as DDIMs. We have showcased the efficacy of TSMs and IR in minimizing the disparity between DDPMs and deterministic baselines across a diverse set of experiments, assessing various facets of our approaches. Our proposed methods significantly enhance sampling speed and quality in fluid dynamics simulations and we posit their potential applicability in other domains for which diffusion models are considered state-of-the-art.

**Future work.** For TSMs, transfer learning is a desirable alternative to aid in reducing the computational burden of training from scratch. Additionally, integrating TSMs and IR together and with other enhancement methods holds promise in further improving the accuracy of DMs, relying on their flexibility for seamless integration with various techniques such as guidance (Dhariwal and Nichol, 2021; Ho and Salimans, 2021; Hong et al., 2023), distillation (Luhman and Luhman, 2021; Salimans and Ho, 2022), and latent diffusion models (Rombach et al., 2022). Finally, through our proposed formulation of the spatio-temporal prediction task, flexible sampling with novel sampling schemes (similar to Harvey et al. (2022)) is another interesting venue of future research.

ACKNOWLEDGMENTS

The authors would like to thank Björn List for his valuable comments and the insightful discussions during the early stages of this work. We also express our gratitude to Georg Kohl and Qiang Liu for their support and detailed clarifications regarding the `Tra` and `Air` test cases.

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

## A EXPERIMENTAL SETUP

We summarize the details for all our datasets in Table 3. For detailed information regarding the generation of these datasets, please refer to the corresponding papers for `Tra` (Kohl et al., 2024) and `Air` (Liu and Thuerey, 2024). For the `Tra` and `Fturb` cases, only a single parameter is varied, namely $Re$, and is provided as an input channel to all models. While in `Tra` we consider velocity ($u_x$ and $u_y$), pressure ($p$), and density ($\rho$) fields as input and output fields, the velocity fields are only considered in `Fturb`. Additionally, for both datasets, the models are conditioned on the prediction stride $j$, provided as an additional input channel to the network.

In the `Air` case, there are three main parameters being changed: airfoil shape $\mathcal{S}$, angle of attack (AoA), and $Re$. These parameters are provided as three input fields/channels to the network and the model outputs the velocity and pressure dimensionless fields. The number of outputs obtained from the different solver settings (referred to as "Sequences per Param" in Table 3) are fixed to 25 solutions for every parameter configuration. For the `Air`$_{\text{One}}$ case, $\mathcal{S}$ is fixed to the *raf30* airfoil shape and an AoA of 20°, while in `Air`$_{\text{Multi}}$, the three parameters are varied.

**Generating the `Fturb` dataset.** We solve the incompressible form of the NS PDE (Eq. 4 with $\partial\rho/\partial\tau = 0$) using a spatially varying sinusoidal forcing term $f = sin(4y)\hat{i} - 0.1u$ (similar to Kochkov et al. (2021)) for a range of $Re$ values for training and testing. Using $\Phi_{\text{flow}}$ (Holl et al., 2020), we discretize a square domain $\Omega = [0, 2\pi] \times [0, 2\pi]$ on a $128 \times 128$ cartesian grid. Both spatial derivative terms (i.e., the advection and diffusion terms) are discretized using a $6^{\text{th}}$-order implicit finite difference method and are solved iteratively using the Conjugate Gradient (CG) solver. The time advancement scheme uses a $4^{\text{th}}$-order Runge–Kutta scheme, which also handles the pressure gradient term with $4^{\text{th}}$-order accuracy using the CG solver. In addition, this step ensures a divergence-free flow velocity. Each RK-4 step consists of 25 iterations to ensure convergence at each timestep. The outputs are then downsampled to $64 \times 64$ resolution. To augment the difficulty of the problem, we record each snapshot after 25 timesteps from the solver (i.e., we consider a temporal stride of 25 frames) with CFL $= 0.7$ and variable $\delta\tau$. Details regarding the number of sequences generated per $Re$ and the total number of frames used for training and testing can be found in Table 3. The only difference between the sequences for each $Re$ is the initial condition, defined as a random noise based on a distinct random seed. To ensure fully convergent flow fields, each simulation is run for 20 timesteps (including the temporal stride) as a warmup before the outputs are recorded.

Table 3: Parameter values for all datasets. *Using 1387 airfoil shapes. **Using 30 airfoil shapes.

| Dataset | Param | Purpose | Values | Sequences per Param | Total Sequences | R. | Total Frames |
|---|---|---|---|---|---|---|---|
| `Tra` ($128 \times 64$) | Ma | training | $\{0.53, 0.54, \ldots, 0.63\} \cup \{0.69, 0.70, \ldots, 0.90\}$ | 1 | 33 | 501 | 16533 |
| | | test | ext: $\{0.50, 0.51, 0.52\}$ | 2 | 6 | 60 | 360 |
| | | | int: $\{0.66, 0.67, 0.68\}$ | 2 | 6 | 60 | 360 |
| | | | long: $\{0.64, 0.65\}$ | 2 | 4 | 240 | 960 |
| `Fturb` ($64 \times 64$) | Re | training | $\{200, 1000, 2500, 4000\}$ | 240 | 960 | 51 | 48960 |
| | | test | ext: $\{100, 5000\}$ | 2 | 4 | 30 | 120 |
| | | | int: $\{1750\}$ | 2 | 2 | 30 | 60 |
| | | | long: $\{100, 1750, 5000\}$ | 2 | 6 | 120 | 720 |
| `Air`$_{\text{One}}$ ($32 \times 32$) | Re | training | $\{1.5{\times}10^6, 3.5{\times}10^6, 5.5{\times}10^6, 7.5{\times}10^6, 9.5{\times}10^6\}$ | 25 | - | - | 125 |
| | | test | ext: $\{5{\times}10^5, 10.5{\times}10^6\}$ | 25 | - | - | 50 |
| | | | int: $\{2.5{\times}10^6, 4.5{\times}10^6, 6.5{\times}10^6, 8.5{\times}10^6\}$ | 25 | - | - | 100 |
| `Air`$_{\text{Multi}}$ ($32 \times 32$) | Re & AoA & $\mathcal{S}$ | training* | Re: $(10^6, 10^7)$, AoA: $(-22.5, 22.5)$ | 25 | - | - | 31250 |
| | | test** | Re: $(5{\times}10^5, 10^6) \cup (10^7, 1.1{\times}10^7)$ AoA: $(-25, -22.5) \cup (22.5, 25)$ | 25 | - | - | 3250 |

## B REFINEMENT SCHEDULES FOR IR

As mentioned in Section 4.2, optimization of the IR schedules $\gamma$ is limited to a straightforward greedy optimization, presented in Algorithm 3. We start the optimization by choosing the max possible length of $\gamma$ ($N$) and a value for the first refinement step ($K$). If $K$ is not provided, the default starting value is $T$ (i.e., the number of noise steps of the pre-trained DDPM). For the first step, we consider $K$ possible options for refinement, i.e., $\{x \in \mathbb{Z} \mid 1 \le x \le K\}$. As long as the validation loss $L$ is decreasing, we keep looping over all $K$ values. However, if the loss does not improve after $tol$ steps (i.e., a tolerance set by the user to achieve early stopping for non-promising optimizations), we

continue to the next refinement step. The starting point for the next timestep is defined to be one step smaller the current optimized step to further reduce the computational cost. If for the current step no value was found to reduce the current best loss $L_{\text{best}}$, the entire optimization is terminated, leading to the loss in Eq. 6, even if $|\gamma| < N$. These restrictions are imposed to reduce the computational cost of optimization, though they might lead to non-optimal results. More expensive gradient-based and gradient-free optimization algorithms offer the potential to yield better results; however, we believe that the accuracy gain would not compensate for the concomitant computational cost.

---

**Algorithm 3** Greedy optimization of $\gamma$ with early stopping

---

**Require:** $N$ (Max length of $\gamma$), $K$ (starting value)
**Require:** $tol$ (tolerance value), $x_{\text{init}}$ (Initial state, $\epsilon_\theta$ (pre-trained DDPM)
1: $\gamma \leftarrow \{\}$
2: **for** $i = 1$ to $N$ **do**
3:     $L_{\text{best}} \leftarrow \infty$
4:     $t_{\text{opt}} \leftarrow -1$
5:     counter $\leftarrow 0$
6:     **for** $j = K$ to $1$ **do**                                           ▷ Go over all possible values.
7:         $\gamma_{\text{temp}} \leftarrow$ append$(\gamma, j)$     ▷ Update $\gamma_{\text{temp}}$ by appending step $j$ to $\gamma$ (unchanged).
8:         $L =$ recursive_sampling$(x_{\text{init}}, \gamma_{\text{temp}}, \epsilon_\theta)$     ▷ Evaluate $\gamma_{\text{temp}}$ on the validation dataset.
9:         **if** $L < L_{\text{best}}$ **then**                 ▷ Update the current step if accuracy is higher.
10:             $L_{\text{best}} \leftarrow L$
11:             $t_{\text{opt}} \leftarrow j$
12:             counter $\leftarrow 0$
13:         **else**
14:             counter $\leftarrow$ counter $+1$
15:             **if** counter $> tol$ **then**
16:                 **break**     ▷ Terminate current step: failed to improve $L_{\text{best}}$ after $tol$ trials.
17:             **end if**
18:         **end if**
19:     **end for**
20:     **if** $t_{\text{opt}} = -1$ **then**
21:         **break**                 ▷ Terminate optimization: no improvement found.
22:     **end if**
23:     append$(\gamma, t_{\text{opt}})$
24:     $K \leftarrow t_{\text{opt}} - 1$             ▷ Consider only smaller values for the next refinement step.
25: **end for**
26: **return** $\gamma$

---

Using the greedy algorithm, the optimization schedules utilized in all our experiments are summarized as follows (and visualized in Figure 5):

$$
\begin{aligned}
\gamma_{1,i} &= 0.805 - 0.2i, & \forall i = 0, 1, 2, 3, 4, \\
\gamma_{2,i} &= 0.805 - 0.1i, & \forall i = 0, 1, 2, \ldots, 8, \\
\gamma_{3,i} &= 0.655 - 0.05i, & \forall i = 0, 1, 2, \ldots, 13, \\
\gamma_{4,i} &= 0.905 - 0.05i, & \forall i = 0, 1, 2, \ldots, 18.
\end{aligned}
\tag{8}
$$

Also, $\gamma_5 = \{1/T\}$. When no $\gamma$ schedule is provided, we use a linear sampling schedule by default:

$$
\gamma_{\text{linear}}(N) = \{x \in \mathbb{N} \mid 0 \leq x < T, \text{for } N \in \mathbb{N}\}.
\tag{9}
$$

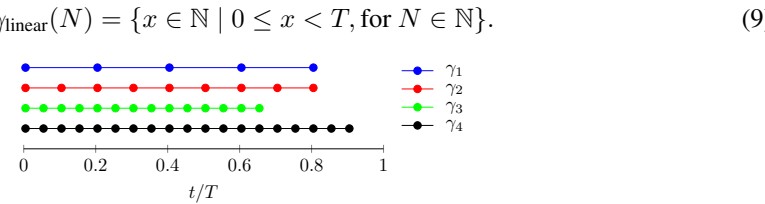

Figure 5: Illustration for the refinement schedules defined in Equation 8. Each node represents a noise step. All schedules start from the node with the largest $t/T$.

## C    SIMULATION PARAMETERS

The training hyperparameters used for all test cases are presented in Table 4. Our training is limited to the L2 loss for DDPMs, TSMs, EDMs and any baselines. We employ an overlap by one frame (see Appendix F for details regarding dataset overlapping) for the `Fturb` case, except for the case of $s = 1$ in TSMs where we overlap 6 frames, since the training procedure was unstable for the first few epochs, leading to suboptimal results. However, we stop the training at half the number of epochs reported in Table 4 to maintain the same number of training iterations for a fair comparison. In `Tra`, all models were trained with max overlap (i.e., 10 frames overlapping) but with early stop once the validation loss stabilizes, typically much earlier than the number of epochs presented in Table 4.

Table 4: Summary of the training hyperparameters employed in all test cases.

| Parameter | Fturb | Tra | Air$_{One}$ | Air$_{Multi}$ |
|---|---|---|---|---|
| Data size (frames) | 48960 | 16533 | 150 | 31250 |
| Resolution | 64×64 | 128×64 | 32×32 | 32×32 |
| Batch size | 64 | 32 | 25 | 128 |
| Epochs | 300 | 3000 | 12000 | 3000 |
| Learning rate (start, end) | $10^{-4}, 10^{-5}$ | $10^{-4}$ | $10^{-4}, 10^{-5}$ | $10^{-4}, 10^{-5}$ |
| Learning rate schedule | Cosine | None | Cosine | Cosine |
| Optimizer | AdamW | AdamW | AdamW | AdamW |
| Weight decay | $10^{-2}$ | $10^{-2}$ | $10^{-2}$ | $10^{-2}$ |
| EMA decay | 0.999 | 0.999 | 0.999 | 0.999 |
| Early stop? | No | Yes | Yes | No |

The utilized network architectures and the diffusion-related hyperparameters are summarized in Table 5. We use the exact same network architectures as provided in benchmark papers for fair comparison and to demonstrate the applicability of our approaches on diverse settings and architectures. For the baseline method trained in the `Fturb` case, we use $c_{in} = 4$ instead of 6 as the state variable $x_t$ is ignored and the network is not conditioned on the noise step $t$. Also, the baseline is trained to directly predict the fluid state $x_{\tau+j\delta\tau}$ instead of the noise $\epsilon_\theta$ as in DDPMs. Apart from these modifications, the training of the baseline and DDPMs is identical.

**EDM training and sampling.**    Our implementation of EDMs is based on the work by Karras et al. (2022). We implement their Algorithm 1 (deterministic sampler) and Algorithm 2 (stochastic sampler) using 1$^{st}$-order Euler and 2$^{nd}$-order Heun's methods with design choices from the last column of Table 1 (Karras et al., 2022). Additionally, we consider preconditioning and a weighted loss function using the parameters recommended by the authors. For each case, we ran the model using different combination of deterministic/stochastic samplers and Euler's/Heun's methods. For the stochastic sampler, the parameters $\{S_{churn}, S_{tmin}, S_{tmax}, S_{noise}\}$ were non-comprehensively tuned to attain the best possible results. We found the values 10, 0, $\infty$, and 1 for $S_{churn}, S_{tmin}, S_{tmax}$, and $S_{noise}$, respectively, to generally yield satisfactory results across datasets. Noteworthy, EDMs introduce more hyperparameters for tuning compared to DDPMs, DDIMs, and our approaches.

Table 5: Network architecture and diffusion-related hyperparameters used in all test cases. $c_{in}$ and $c_{out}$ refer to the network's number of input and output channels, respectively. $\beta_{start}$ and $\beta_{end}$ of the noise schedule $\beta_t$ are similar to Ho et al. (2020) but are scaled by a factor depending on the chosen noise steps T, as defined in Nichol and Dhariwal (2021) and Kohl et al. (2024).

| Parameter | Fturb | Tra | Air$_{One}$ | Air$_{Multi}$ |
|---|---|---|---|---|
| Architecture | Kohl et al. (2024) | Kohl et al. (2024) | Liu and Thuerey (2024) | Liu and Thuerey (2024) |
| $c_{in}$ | 6 | 10 | 6 | 6 |
| $c_{out}$ | 2 | 4 | 3 | 3 |
| $\beta_{start}$ | $10^{-4} \cdot (400/T)$ | $10^{-4} \cdot (500/T)$ | $10^{-4} \cdot (1000/T)$ | $10^{-4} \cdot (1000/T)$ |
| $\beta_{end}$ | $0.02 \cdot (400/T)$ | $0.02 \cdot (500/T)$ | $0.02 \cdot (1000/T)$ | $0.02 \cdot (1000/T)$ |
| Schedule | Linear | Linear | Linear / Cosine | Linear |

## D    Parallel recursive sampling

**Speedup through parallel sampling.**    One of the benefits of conditioning a surrogate model on the prediction stride $j$ is to enable parallel sampling of transient test cases. To date, related work concerned with physics-based simulations of transient nature focus on autoregressive sampling only (refer to Section 3 for more details). Nonetheless, we posit that our formulation of the problem definition (see Section 2.2), combined with our proposed improvement approaches (i.e., TSMs and IR), would enable parallel sampling with reduced inference time for the same computational budget.

Inspired by the progress in video diffusion models (Ho et al.; 2022; Ni et al., 2023; Wu et al., 2023; Harvey et al., 2022), we present two examples of sampling schemes that allow for parallel sampling of diffusion models in transient test cases. As demonstrated in Fig. 6 for a sample problem with $R = 10$ states, while autoregressive sampling (see Fig. 6a) requires 10 successive predictions to sample the entire simulation, it only requires 5 steps for parallel sampling with a batch size $n = 2$ (cf. Fig. 6b). With $n = 5$, max parallelism can be achieved to sample the entire simulation in 2 steps only (cf. Fig. 6c). This corresponds to a speedup factor $\approx n$, assuming that $\mathcal{T} \geq n$. For the two parallel sampling schemes, we assume that sampling with $j = 5$ will lead to lower TA-MSE compared to sampling with $j = 1$ for the entire simulation.

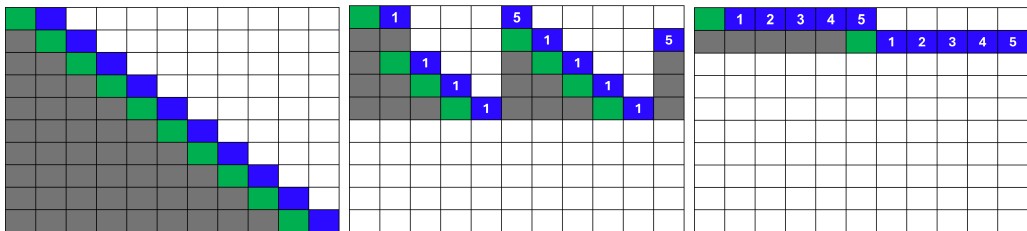

(a) Autoregressive sampling ($n = 1$).    (b) Parallel sampling with $n = 2$.    (c) Max parallelism with $n = 5$.

Figure 6: Parallel sampling of a time-dependent simulation with $R = 10$, enabled through models conditioned on the prediction stride $j$. $n$ is the batch size used for sampling. Each row represents the state of the entire simulation, where the states evolve in time going downwards through the rows. Numbers within blue cells (cells to predict) correspond to the value of $j$ used to sample from the nearest initial condition (green cell). Gray cells are states already sampled and are not needed for current or subsequent predictions. For (b) and (c), we assume that $j = 5$ leads to highest prediction accuracy compared to other $j$ values.

**Higher accuracy when sampling intermediate timesteps.**    For models that allow sampling using $j > 1$, we not only enable flexible parallel sampling to reduce inference time as discussed, but also we improve the overall prediction accuracy. Typically when intermediate states are sampled with a stride $j_{\text{sec}}$ smaller than the primary sampling stride $j$, we gain (marginally) higher overall accuracy compared to ignoring intermediate states for sampling (using $j$ only). In Table 6, we demonstrate this by sampling the best IR and TSM models for the `Tra` and `Fturb` test cases. In both test cases, we see marginal improvement in the mean TA-MSE for small $j$; however, the improvement is more pronounced for higher $j$ values. This is primarily attributed to the argument discussed by Lienen et al. (2024) pertaining to higher accuracy of predicting with small timestep sizes for a short rollout horizon ($R = j$ for sampling the intermediate states) before error accumulation predominates, leading to lower accuracy for longer rollout horizons. This greatly motivates future research in flexible sampling of time-dependent physics-based simulations similar to Harvey et al. (2022).

Table 6: Mean TA-MSE results for transient cases for predictions with $j > 1$, and with and without sampling the intermediate states. For the former, the model names include "**(all)**". TA-MSE is averaged over `ext` and `int` regions. Standard deviation values are calculated over all timesteps, both regions, and multiple samples. The results show consistent improvement when sampling the intermediate states with $j_{\text{sec}} = 1$ over sampling with the primary prediction stride $j$ only.

| | $j$ | | | | |
| Model | 2 | 4 | 6 | 8 | 10 |
| --- | --- | --- | --- | --- | --- |
| | `Tra` $(10^{-3})$ | | | | |
| IR T100 - $\mathcal{N}$ $\gamma_1$ | $2.06 \pm 1.91$ | $2.73 \pm 2.61$ | $3.15 \pm 2.93$ | $4.18 \pm 3.39$ | $3.54 \pm 2.98$ |
| IR T100 - $\mathcal{N}$ $\gamma_1$ **(all)** | $1.94 \pm 1.82$ | $2.33 \pm 2.27$ | $2.75 \pm 2.83$ | $3.06 \pm 2.82$ | $2.72 \pm 3.39$ |
| TSM T100 $s = 1$ | $1.46 \pm 1.55$ | $1.78 \pm 1.82$ | $2.17 \pm 2.11$ | $2.35 \pm 2.07$ | $2.32 \pm 2.12$ |
| TSM T100 $s = 1$ **(all)** | $1.36 \pm 1.47$ | $1.59 \pm 1.68$ | $1.93 \pm 2.22$ | $1.74 \pm 1.74$ | $2.09 \pm 3.22$ |
| | `Fturb` $(10^{-2})$ | | | | |
| IR T80 - $\mathcal{N}$ $\gamma_4$ | $2.18 \pm 0.88$ | $2.97 \pm 0.95$ | $2.28 \pm 0.81$ | $4.18 \pm 1.91$ | $2.97 \pm 0.55$ |
| IR T80 - $\mathcal{N}$ $\gamma_4$ **(all)** | $1.89 \pm 0.77$ | $2.09 \pm 0.75$ | $2.58 \pm 3.40$ | $3.36 \pm 4.23$ | $3.47 \pm 6.47$ |
| TSM T80 $s = 0.75$ | $3.63 \pm 1.95$ | $5.46 \pm 2.57$ | $6.61 \pm 2.84$ | $10.14 \pm 4.42$ | $5.48 \pm 0.62$ |
| TSM T80 $s = 0.75$ **(all)** | $3.22 \pm 1.77$ | $4.07 \pm 2.17$ | $5.10 \pm 3.29$ | $6.19 \pm 4.62$ | $4.79 \pm 7.15$ |

# E    FULL RESULTS OF EXPERIMENTS

In this section, we present our full results from all test cases in addition to the top performing models presented in Section 5. The results are split into quantitative and qualitative sections.

## E.1    QUANTITATIVE RESULTS

**`Tra` test case**    The results for using $j = 1$ and optimal $j > 1$ are presented in Table 7. For TA-MSE values obtained with $j = 1$, several insights can be observed. First, in comparison to ACDM (Kohl et al., 2024), we are able to provide more accurate results through IR and TSMs with single-step and few-step sampling (NFEs $\leq 5$), even though our problem formulation (defined in Section 2) is much harder compared to ACDM for the same utilized architecture. Noteworthy, the ACDM results are obtained by conditioning over two previous timesteps, whereas all our models are conditioned on the previous timestep only. Second, various TSMs are capable of improving over all considered baselines with and without advanced training mechanisms, with the most accurate model being a single-step solver. IR only improves over the base DDPMs and surpasses DDIMs by a significant margin for both similar and optimized sampling schedules. In our experiments, we focused on using random Gaussian noise $\mathcal{N}(0, \mathbf{I})$ for the initial state $x_{\text{init}}$ to reduce the computational overhead before starting the refinement process and also because we found in early experiments that this setup yields slightly more accurate results than starting from a prediction obtained through truncated sampling of the base DDPM. Third, IR and DDIM were able to provide noise-free results even when the base DDPM produces noisy output (e.g. DDPM T20). Fourth, in comparison to EDMs, we observe that although using a deterministic Euler solver yields the best results for this case, with a negligible margin over TSM, it requires 4 NFEs. In contrast, TSM is the only model that enables single-step, high-fidelity inference. Note that we evaluate several solvers for EDM (see Fig. 7, left), but we only report the optimal settings for each solver in Table 7. Finally, we note that PDE-Refiner performs suboptimally with a significant gap compared to other baselines. The model was found to be highly sensitive to its key hyperparameters (Kohl et al., 2024), rendering the hyperparameter tuning for this model challenging and ultimately resulting in suboptimal performance.

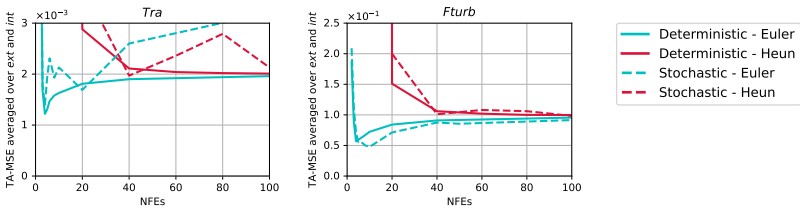

Figure 7: Evaluating different combinations of EDM samplers for the `Tra` (left) and `Fturb` (right) test cases. Standard deviation regions are omitted for clarity.

For $j > 1$, we observe substantial improvements in accuracy, even for low-fidelity models, except for EDMs. Suboptimal few-steps DDIMs are able to achieve comparable accuracy to the best baselines, which is also true for IR and TSMs. Noteworthy, the numbers reported for $j > 1$ are limited to sampling with $j$ only without considering the intermediate states. We show in Appendix D that sampling intermediate states could further enhance the overall accuracy, particularly for large $j$.

Table 7: TA-MSE $(10^{-3})$ values of the `Tra` test case for $j = 1$ (left) and the best results using the optimum $j$ (right). $UNet_{tn}$ and $UNet_{ut}$ refer to a UNet with training noise and unrolled training, respectively. The reader is referred to Kohl et al. (2024) for a full description of the baselines reported. Lowest TA-MSE values for each group of models are highlighted in **bold**. Standard deviation values are obtained over all timesteps and multiple samples. *Model produces noisy output.

| | | $j = 1$ | | | optimum $j$ | |
| --- | --- | --- | --- | --- | --- | --- |
| Model | NFEs | ext | int | $j$ | ext | int |
| **Kohl et al. (2024)** | | | | | | |
| ACDM | 20 | $2.3 \pm 1.4$ | $2.7 \pm 2.1$ | | | |
| $UNet_{tn}$ | 1 | $\mathbf{1.4 \pm 0.8}$ | $1.8 \pm 1.1$ | | | |
| $UNet_{ut}$ | 1 | $1.6 \pm 0.7$ | $\mathbf{1.5 \pm 1.5}$ | | | |
| $ResNet_{dil.}$ | 1 | $1.7 \pm 1.0$ | $1.7 \pm 1.4$ | | | |
| $FNO_{16}$ | 1 | $4.8 \pm 1.2$ | $5.5 \pm 2.6$ | | | |
| $TF_{Enc}$ | 1 | $3.3 \pm 1.2$ | $6.2 \pm 4.2$ | | | |
| PDE-Refiner | 4 | $5.4 \pm 2.1$ | $7.1 \pm 2.1$ | | | |
| **DDPM** | | | | | | |
| T20* | 20 | $3.5 \pm 3.1$ | $\mathbf{2.6 \pm 2.1}$ | 4 | $\mathbf{1.6 \pm 1.3}$ | $\mathbf{2.1 \pm 2.0}$ |
| T100 | 100 | $\mathbf{3.0 \pm 2.7}$ | $4.1 \pm 3.7$ | 5 | $1.9 \pm 1.8$ | $3.1 \pm 2.7$ |
| **EDM** | | | | | | |
| Euler - Deterministic | 4 | $\mathbf{1.3 \pm 1.3}$ | $\mathbf{1.1 \pm 1.0}$ | 1 | $\mathbf{1.3 \pm 1.3}$ | $\mathbf{1.1 \pm 1.0}$ |
| Stochastic | 4 | $1.7 \pm 1.7$ | $\mathbf{1.1 \pm 1.0}$ | 1 | $1.7 \pm 1.7$ | $\mathbf{1.1 \pm 1.0}$ |
| Heun - Deterministic | 40 | $2.7 \pm 2.7$ | $1.6 \pm 1.4$ | 2 | $2.2 \pm 2.1$ | $\mathbf{1.2 \pm 1.1}$ |
| Stochastic | 40 | $2.0 \pm 2.2$ | $2.0 \pm 1.8$ | 2 | $1.8 \pm 2.0$ | $1.8 \pm 1.6$ |
| **DDIM** | | | | | | |
| T20 | 5 | $4.4 \pm 3.3$ | $6.7 \pm 5.4$ | 4 | $1.9 \pm 1.7$ | $2.5 \pm 2.2$ |
| | 10 | $\mathbf{3.2 \pm 2.7}$ | $\mathbf{4.2 \pm 3.9}$ | 4 | $\mathbf{1.4 \pm 1.3}$ | $\mathbf{1.8 \pm 1.7}$ |
| T100 | 5 | $7.8 \pm 5.2$ | $6.1 \pm 4.3$ | 10 | $2.8 \pm 1.9$ | $4.1 \pm 3.0$ |
| | 10 | $8.8 \pm 5.5$ | $6.7 \pm 4.3$ | 10 | $3.3 \pm 2.2$ | $4.5 \pm 3.0$ |
| | 20 | $7.8 \pm 5.2$ | $6.1 \pm 4.3$ | 10 | $2.8 \pm 1.9$ | $4.1 \pm 3.0$ |
| | 50 | $7.1 \pm 4.7$ | $6.0 \pm 4.4$ | 9 | $3.2 \pm 2.2$ | $3.3 \pm 2.3$ |
| **IR (Ours)** | | | | | | |
| T20 | 5 | $2.8 \pm 2.9$ | $1.7 \pm 1.7$ | 4 | $1.3 \pm 1.4$ | $1.5 \pm 1.5$ |
| | 10 | $3.2 \pm 2.9$ | $1.6 \pm 1.5$ | 4 | $\mathbf{1.2 \pm 1.1}$ | $1.6 \pm 1.6$ |
| T100 | 5 | $1.7 \pm 1.4$ | $2.1 \pm 1.8$ | 1 | $1.7 \pm 1.4$ | $2.1 \pm 1.8$ |
| | 10 | $3.8 \pm 3.2$ | $3.5 \pm 3.3$ | 2 | $2.6 \pm 2.3$ | $2.7 \pm 2.4$ |
| | 20 | $3.1 \pm 3.1$ | $3.0 \pm 2.9$ | 5 | $1.9 \pm 1.7$ | $3.0 \pm 2.8$ |
| | 50 | $4.4 \pm 4.0$ | $3.5 \pm 3.7$ | 5 | $1.7 \pm 1.5$ | $4.1 \pm 3.8$ |
| T20, $\gamma_1$ | 5 | $2.3 \pm 2.3$ | $2.0 \pm 1.9$ | 4 | $1.3 \pm 1.4$ | $1.7 \pm 1.8$ |
| $\gamma_2$ | 9 | $3.1 \pm 2.9$ | $1.6 \pm 1.6$ | 4 | $1.3 \pm 1.2$ | $1.7 \pm 1.7$ |
| $\gamma_3$ | 14 | $1.7 \pm 1.8$ | $1.7 \pm 1.8$ | 3 | $1.7 \pm 1.7$ | $1.5 \pm 1.6$ |
| $\gamma_4$ | 19 | $2.4 \pm 2.6$ | $\mathbf{1.2 \pm 1.1}$ | 3 | $1.3 \pm 1.3$ | $\mathbf{1.4 \pm 1.4}$ |
| T100, $\gamma_1$ | 5 | $\mathbf{1.6 \pm 1.3}$ | $2.0 \pm 1.7$ | 1 | $1.6 \pm 1.3$ | $2.0 \pm 1.7$ |
| $\gamma_2$ | 9 | $3.7 \pm 3.0$ | $2.7 \pm 2.4$ | 2 | $2.7 \pm 2.4$ | $2.7 \pm 2.4$ |
| $\gamma_3$ | 14 | $3.2 \pm 3.0$ | $2.9 \pm 2.8$ | 2 | $2.5 \pm 2.8$ | $2.7 \pm 2.5$ |
| $\gamma_4$ | 19 | $2.6 \pm 2.6$ | $3.1 \pm 2.9$ | 2 | $2.2 \pm 2.4$ | $2.6 \pm 2.3$ |
| **TSM (Ours)** | | | | | | |
| T15, $s = 0.25$ | 11 | $1.8 \pm 1.8$ | $1.7 \pm 1.5$ | 2 | $\mathbf{1.2 \pm 1.2}$ | $\mathbf{1.5 \pm 1.6}$ |
| $s = 0.5$ | 7 | $2.1 \pm 2.2$ | $\mathbf{1.2 \pm 1.1}$ | 3 | $\mathbf{1.2 \pm 1.1}$ | $1.6 \pm 1.4$ |
| $s = 0.75$ | 3 | $1.4 \pm 1.4$ | $2.4 \pm 2.3$ | 6 | $1.4 \pm 1.1$ | $1.6 \pm 1.4$ |
| $s = 1$ | 1 | $1.3 \pm 1.2$ | $1.6 \pm 1.5$ | 2 | $\mathbf{1.2 \pm 1.1}$ | $1.6 \pm 1.6$ |
| T100, $s = 1$ | 1 | $\mathbf{1.2 \pm 1.1}$ | $1.5 \pm 1.5$ | 1 | $\mathbf{1.2 \pm 1.1}$ | $\mathbf{1.5 \pm 1.5}$ |

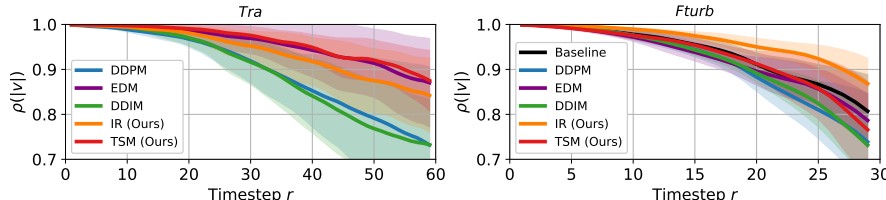

Figure 8: `Tra` (left) and `Fturb` (right) absolute velocity $|u|$ correlation to ground truth using $j = 1$ for the models reported in Table 1. Values are averaged over `ext` and `int` regions and the shaded area represents the standard deviation from both regions and multiple samples.

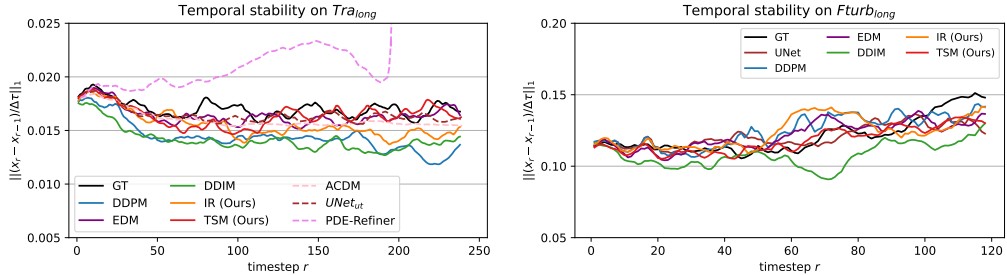

Figure 9: Time evolution for the temporal stability parameter between consecutive timesteps for top-performing models (see Table 1) compared to the ground truth simulation (GT). Dashed lines are obtained from Kohl et al. (2024). Standard deviation regions are omitted for clarity.

**Temporal analysis.**    In Fig. 8 (left), we see how the $|u|$ correlation to the ground truth evolves over time for the most accurate models, demonstrating consistency with the TA-MSE metric. In addition, we evaluate the temporal stability of these top-performing models on the $\texttt{Tra}_{long}$ dataset, which has much longer sequence length (i.e., $R = 240$ instead of 60), to ensure that the models' predictions remain physically consistent even after the predicted trajectories have significantly deviated from the ground truth. In Fig. 9 (left), we compare the time evolution of the stability parameter, defined as the rate of change of each flow field $\mathbf{x}$, by evaluating $||(\mathbf{x}_r - \mathbf{x}_{r-1})/\Delta\tau||_1$. Compared to the ground truth results, we see that stability parameter remains bounded within the range $[0.015, 0.02]$ for both the ground truth and all models, except for a low-fidelity PDE-Refiner model that produces non-physical predictions beyond $r \approx 200$. While DDIM, DDPM, and IR models slightly exceed the lower bound, they maintain temporal stability.

**`Fturb` test case**    The results for using $j = 1$ and $j > 1$ are presented in Table 8. Similar to the previous test case with $j = 1$, we observe comparable trends in accuracy for the different models. IR sampling with DDPM T80 base model provides the most accurate results with NFEs as low as 5 steps that surpass the baseline with a significant margin. Best result is achieved by a 19-step IR model with optimized schedule $\gamma_4$. However, with much less NFEs, highly satisfactory results are possible, such as using $\gamma_1$ for the DDPM T80 base model. Single-step TSMs also provide competitive accuracy values which are marginally below the accuracy of the baseline. By using optimum $j$ values, single-step TSMs are able to surpass the baseline and provide comparable results to IR, which provides further better results than with $j = 1$. The most accurate EDM is achieved through a stochastic Euler solver with 10 NFEs, delivering much better results than all 2nd-order Heun solvers with a fraction of NFEs. The model has an optimum $j = 1$; hence, it is outperformed by most DDPMs, DDIMs, IR, and TSMs at lower NFEs when the optimum $j > 1$. Although the most accurate EDM solver is on par with the single-step TSM in terms of accuracy, it is $10\times$ slower and doesn't provide better results when $j > 1$.

**Temporal analysis.**    The time evolution of $\rho(|u|)$ is shown in Fig. 8 (right), while the stability parameter for long rollouts evaluated on $\texttt{Fturb}_{long}$ with $R = 120$ instead of 30 is presented in Fig. 9 (right). Both figures show consistent results to the TA-MSE values and demonstrate the accuracy of the different models over time. Noteworthy, the stability parameter (Fig. 9, right) of the

ground truth and all models remains constrained within $[0.1, 0.15]$, with the exception of a minor deviation observed for DDIM at $r \approx 70$, which doesn't lead to non-physical results for the remaining timesteps.

Table 8: TA-MSE $(10^{-2})$ values of the `Fturb` test case for $j = 1$ (left) and the best results using the optimum $j$ (right). Lowest TA-MSE values for each group of models are highlighted in **bold**. Standard deviation values are obtained over all timesteps and multiple samples. *Model produces noisy output.

| Model | NFEs | $j = 1$ ext | $j = 1$ int | $j$ | optimum $j$ ext | optimum $j$ int |
|---|---|---|---|---|---|---|
| Baseline | 1 | $3.95 \pm 3.84$ | $4.82 \pm 5.23$ | 3 | $3.22 \pm 2.88$ | $3.73 \pm 3.55$ |
| **DDPM** | | | | | | |
| T20* | 20 | $20.66 \pm 16.71$ | $24.01 \pm 20.58$ | 10 | $7.80 \pm 5.22$ | $4.51 \pm 3.66$ |
| T50 | 50 | $10.60 \pm 11.47$ | $9.85 \pm 8.76$ | 3 | $7.31 \pm 7.28$ | $5.54 \pm 5.07$ |
| T80 | 80 | $\mathbf{6.16 \pm 6.54}$ | $\mathbf{5.27 \pm 5.55}$ | 5 | $\mathbf{4.68 \pm 4.52}$ | $\mathbf{2.71 \pm 2.22}$ |
| **EDM** | | | | | | |
| Euler - Deterministic | 4 | $5.65 \pm 5.58$ | $5.67 \pm 5.77$ | 1 | $5.65 \pm 5.58$ | $5.67 \pm 5.77$ |
| Stochastic | 10 | $\mathbf{4.75 \pm 4.53}$ | $\mathbf{4.78 \pm 4.26}$ | 1 | $\mathbf{4.75 \pm 4.53}$ | $\mathbf{4.78 \pm 4.26}$ |
| Heun - Deterministic | 40 | $9.22 \pm 8.93$ | $12.06 \pm 11.43$ | 5 | $9.23 \pm 9.10$ | $7.48 \pm 6.24$ |
| Stochastic | 100 | $10.32 \pm 9.60$ | $9.35 \pm 8.90$ | 3 | $8.05 \pm 8.27$ | $8.16 \pm 7.73$ |
| **DDIM** | | | | | | |
| T50 | 5 | $14.21 \pm 12.09$ | $26.65 \pm 20.68$ | 5 | $7.64 \pm 6.62$ | $13.31 \pm 12.80$ |
| | 10 | $8.49 \pm 8.74$ | $14.79 \pm 14.07$ | 4 | $6.53 \pm 5.99$ | $8.26 \pm 7.74$ |
| | 25 | $7.27 \pm 7.93$ | $11.84 \pm 11.47$ | 3 | $5.99 \pm 6.17$ | $7.22 \pm 7.46$ |
| T80 | 5 | $9.77 \pm 8.56$ | $19.91 \pm 16.50$ | 6 | $4.78 \pm 4.68$ | $5.92 \pm 5.37$ |
| | 10 | $5.61 \pm 5.79$ | $10.47 \pm 9.90$ | 6 | $4.19 \pm 4.06$ | $3.99 \pm 3.67$ |
| | 20 | $4.59 \pm 4.89$ | $7.72 \pm 7.46$ | 6 | $\mathbf{4.10 \pm 3.97}$ | $3.59 \pm 3.27$ |
| | 40 | $\mathbf{4.31 \pm 4.62}$ | $6.97 \pm 6.84$ | 6 | $4.13 \pm 3.99$ | $\mathbf{3.53 \pm 3.20}$ |
| **IR (Ours)** | | | | | | |
| T50 | 5 | $4.36 \pm 4.65$ | $4.05 \pm 3.86$ | 3 | $3.44 \pm 3.15$ | $2.89 \pm 2.80$ |
| | 10 | $4.44 \pm 4.99$ | $3.24 \pm 3.06$ | 3 | $3.85 \pm 3.79$ | $2.64 \pm 2.59$ |
| | 25 | $4.24 \pm 4.69$ | $3.52 \pm 3.41$ | 2 | $4.27 \pm 4.54$ | $2.56 \pm 2.34$ |
| T80 | 5 | $3.21 \pm 3.37$ | $2.77 \pm 2.67$ | 6 | $\mathbf{2.89 \pm 2.41}$ | $1.86 \pm 1.58$ |
| | 10 | $2.93 \pm 3.34$ | $\mathbf{1.70 \pm 1.63}$ | 2 | $3.36 \pm 3.68$ | $\mathbf{1.19 \pm 1.18}$ |
| | 20 | $2.61 \pm 2.73$ | $2.27 \pm 2.43$ | 6 | $2.88 \pm 2.51$ | $1.54 \pm 1.22$ |
| | 40 | $3.19 \pm 3.41$ | $2.58 \pm 2.59$ | 2 | $3.63 \pm 3.71$ | $1.36 \pm 1.17$ |
| T50, $\gamma_1$ | 5 | $4.36 \pm 4.65$ | $4.05 \pm 3.86$ | 3 | $3.44 \pm 3.15$ | $2.89 \pm 2.80$ |
| $\gamma_2$ | 9 | $5.31 \pm 6.09$ | $2.92 \pm 2.66$ | 3 | $4.16 \pm 4.15$ | $2.46 \pm 2.36$ |
| $\gamma_3$ | 14 | $5.29 \pm 5.84$ | $4.07 \pm 3.94$ | 3 | $4.08 \pm 4.00$ | $3.32 \pm 3.25$ |
| $\gamma_4$ | 19 | $3.59 \pm 3.89$ | $4.08 \pm 4.22$ | 2 | $3.50 \pm 3.51$ | $3.62 \pm 3.77$ |
| T80, $\gamma_1$ | 5 | $3.21 \pm 3.37$ | $2.77 \pm 2.67$ | 6 | $\mathbf{2.89 \pm 2.41}$ | $1.86 \pm 1.58$ |
| $\gamma_2$ | 9 | $3.21 \pm 3.59$ | $2.24 \pm 2.13$ | 6 | $3.42 \pm 3.07$ | $1.75 \pm 1.52$ |
| $\gamma_3$ | 14 | $2.93 \pm 3.31$ | $2.27 \pm 2.08$ | 6 | $2.93 \pm 2.50$ | $1.87 \pm 1.66$ |
| $\gamma_4$ | 19 | $\mathbf{2.59 \pm 2.82}$ | $1.95 \pm 2.02$ | 2 | $\mathbf{2.90 \pm 2.97}$ | $1.46 \pm 1.45$ |
| **TSM (Ours)** | | | | | | |
| T20, $s = 0.5$ | 10 | $6.55 \pm 7.30$ | $6.64 \pm 6.54$ | 2 | $5.81 \pm 6.30$ | $4.57 \pm 4.42$ |
| $s = 0.75$ | 3 | $7.54 \pm 7.88$ | $8.54 \pm 7.38$ | 3 | $5.85 \pm 5.20$ | $6.09 \pm 5.13$ |
| $s = 1$ | 1 | $9.09 \pm 8.92$ | $10.96 \pm 9.01$ | 3 | $14.19 \pm 9.87$ | $1.36 \pm 1.05$ |
| T50, $s = 0.5$ | 25 | $6.74 \pm 7.61$ | $5.48 \pm 5.39$ | 3 | $4.43 \pm 4.45$ | $5.06 \pm 5.29$ |
| $s = 0.75$ | 13 | $5.10 \pm 4.90$ | $5.94 \pm 6.01$ | 2 | $3.90 \pm 3.67$ | $4.73 \pm 4.56$ |
| $s = 1$ | 1 | $5.47 \pm 6.25$ | $5.94 \pm 6.13$ | 5 | $3.53 \pm 3.25$ | $2.40 \pm 2.37$ |
| T80, $s = 0.5$ | 40 | $5.46 \pm 5.80$ | $3.68 \pm 3.33$ | 3 | $3.89 \pm 3.61$ | $3.58 \pm 3.63$ |
| $s = 0.75$ | 20 | $\mathbf{3.84 \pm 3.85}$ | $\mathbf{4.35 \pm 3.96}$ | 2 | $\mathbf{3.43 \pm 3.03}$ | $3.83 \pm 3.59$ |
| $s = 1$ | 1 | $5.29 \pm 5.33$ | $6.32 \pm 6.69$ | 5 | $4.44 \pm 4.13$ | $1.93 \pm 1.86$ |
| T100, $s = 1$ | 1 | $5.00 \pm 5.52$ | $4.39 \pm 4.76$ | 5 | $3.91 \pm 3.59$ | $\mathbf{1.20 \pm 1.04}$ |

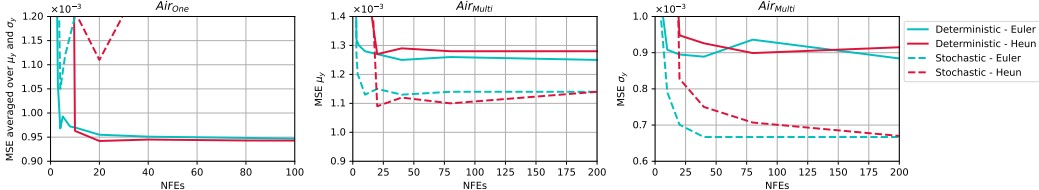

Figure 10: Evaluating different combinations of EDM samplers for the `Air_One` (left) and `Air_Multi` (middle and right) test cases. Standard deviation regions are omitted for clarity.

**Air test case** The results for the `Air_One` and `Air_Multi` test cases are presented in Table 9. For the `Air_One` test case, as discussed in Section 5, we were able to provide better results than the benchmark by Liu and Thuerey (2024) with the same number of NFEs by using a linear $\beta_t$, alongside dataset normalization. Similar to the two previous test cases, DDIMs reduce NFEs by compromising accuracy, while our IR approach leads to the most accurate results for this study. We found that using $x_{\text{init}} \sim \mathcal{N}(0, \mathbf{I})$ similar to the two previous test cases lead to suboptimal results that are not report here. However, relying on the accurate DDPM base models, we provide $x_{\text{init}}$ through truncated sampling of the base model. This truncation of the base DDPM yields adequately accurate results but are dominated by noise in the output. In this case, IR simply refines the input (which is conceptually the essence of the approach) to remove the noise in the output while maintaining or slightly improving the accuracy. Here, only a single step of refinement was sufficient to provide the highest accuracy results. Regarding TSMs, we observe competitive accuracy for both linear and cosine $\beta_t$ with moderate NFEs ($< 20$), which already reduces the computational cost compared to the benchmark and the base DDPM by a significant margin. While for linear $\beta_t$ single step inference is possible to learn the entire data distribution, the output from these models were found to be slightly noisy. For the cosine $\beta_t$, single-step TSM fails to learn the data distribution (as demonstrated by the extremely high $\text{MSE}_\sigma$) and only learns the mean (as shown by the low $\text{MSE}_\mu$). Hence, this leads to the conclusion that with the proper choice of $\beta_t$ and the associated diffusion-related parameters, single-step inference through TSMs is made possible even in test cases with high uncertainty. Additionally, we evaluate several EDM samplers for this test case and conclude that there is little variability between the top-performing solvers in terms of accuracy. The most accurate EDM is a deterministic $2^{\text{nd}}$-order Heun solver that fails to deliver competitive results. Even at higher NFEs, up to 200, we observe only a minor improvement in accuracy (see Fig. 10, left). This outcome is likely attributable to suboptimal hyperparameter configurations for both training and sampling.

Based on these insights, we restrict our experiments for the `Air_Multi` case to few models only, especially that training and sampling are considerably expensive for this study. We, therefore, only train a DDPM with $T = 100$ and TSMs with different $s$ values, excluding single-step inference since it was unsuccessful in the previous study. We see equivalent results in this study: IR sampling achieves the best results by refining a noisy estimation from the base DDPM, followed by TSM with NFEs $= 25$ and 10. DDIMs are the least accurate even with high NFEs. The results of evaluating multiple EDM solvers are presented in Fig. 10 (middle and right). For $(\text{MSE}_{\boldsymbol{\mu}_y})_a$, all samplers, even those with NFEs $\leq 25$, outperform all other models. However, for $(\text{MSE}_{\boldsymbol{\sigma}_y})_a$, achieving satisfactory results compared to the top DDPM, TSM, and IR models requires NFEs $\geq 40$. Nonetheless, the most accurate sampler exhibits lower accuracy than these models.

Note that heteroscedastic models only predict the data uncertainty by assuming a distribution for the data and estimating its parameters. Consequently, samples drawn from this learned distribution are often non-physical (e.g., see Fig. 9 in Liu and Thuerey (2024)). In contrast, DMs reconstruct the target distribution of solutions, which is then used to estimate the uncertainty. Therefore, heteroscedastic models cannot be directly compared to DDPMs in terms of speed. Nonetheless, they are an established method for predicting the data uncertainty in this area and represent a strong baseline for comparison.

Table 9: $\texttt{Air}_{\texttt{One}}$ (left) and $\texttt{Air}_{\texttt{Multi}}$ (right) test case results. Best results in each group of models are highlighted in **bold**. Standard deviation values are obtained from multiple samples. *Model uses cosine $\beta_t$. #Model only predicts $\mu_y$ and $\sigma_y$. **Model produces slightly noisy output. ***Model produces too noisy output.

$\texttt{Air}_{\texttt{One}}$ $(10^{-4})$

| Model | NFEs | $(\text{MSE}_{\boldsymbol{\mu}_y})_a$ | $(\text{MSE}_{\boldsymbol{\sigma}_y})_a$ |
|---|---|---|---|
| **Liu and Thuerey (2024)** | | | |
| DDPM T200C* | 200 | **3.24 ± 0.55** | 7.66 ± 0.74 |
| Heteroscedastic# | 1 | 3.35 ± 0.60 | **4.47 ± 0.37** |
| **DDPM** | | | |
| T100 | 100 | 3.73 ± 0.34 | 7.73 ± 0.24 |
| T200 | 200 | **2.88 ± 0.26** | **7.05 ± 0.22** |
| T200C* | 200 | 23.0 ± 0.39 | 6.28 ± 0.35 |
| **EDM** | | | |
| Euler - Determ. | 20 | 8.28 ± 1.85 | 10.8 ± 1.51 |
| Stoch. | 4 | 9.66 ± 0.95 | 11.3 ± 1.94 |
| Heun - Determ. | 20 | **8.13 ± 1.28** | **10.1 ± 1.67** |
| Stoch. | 20 | 10.4 ± 0.75 | 11.8 ± 0.27 |
| **DDIM** | | | |
| T100 | 10 | 8.42 ± 0.27 | 10.80 ± 0.51 |
| | 25 | 4.46 ± 0.31 | 8.09 ± 0.23 |
| | 50 | 3.81 ± 0.27 | 7.71 ± 0.22 |
| T200 | 10 | 9.12 ± 0.33 | 10.40 ± 0.54 |
| | 25 | 4.67 ± 0.40 | 7.65 ± 0.29 |
| | 50 | 3.98 ± 0.44 | 7.33 ± 0.25 |
| | 100 | **3.68 ± 0.44** | **7.24 ± 0.25** |
| **IR (Ours)** | | | |
| T100, $s = 0.6$, $\gamma_5$ | 41 | 2.87 ± 0.32 | 6.76 ± 0.20 |
| T200, $s = 0.6$, $\gamma_5$ | 81 | **2.07 ± 0.21** | **6.07 ± 0.26** |
| **TSM (Ours)** | | | |
| T100, $s = 0.5$ | 50 | 3.26 ± 0.42 | 7.71 ± 0.20 |
| $s = 0.75$ | 25 | 3.25 ± 0.38 | 6.98 ± 0.18 |
| $s = 0.9$ | 10 | 3.30 ± 0.39 | **5.89 ± 0.33** |
| $s = 0.95$ | 5 | 4.98 ± 0.60 | 8.83 ± 0.42 |
| $s = 1$** | 1 | 4.84 ± 0.13 | 5.96 ± 0.31 |
| T200, $s = 0.5$ | 100 | 3.59 ± 0.31 | 8.16 ± 0.24 |
| $s = 0.75$ | 50 | **3.02 ± 0.39** | 6.99 ± 0.34 |
| $s = 0.9$ | 20 | 3.92 ± 0.40 | 7.08 ± 0.42 |
| $s = 0.95$ | 10 | 4.24 ± 0.23 | 9.08 ± 0.15 |
| $s = 1$** | 1 | 4.58 ± 0.22 | 8.62 ± 0.40 |
| T200C*, $s = 0.25$ | 150 | 5.32 ± 0.58 | 8.73 ± 0.28 |
| $s = 0.5$ | 100 | 3.44 ± 0.30 | 8.03 ± 0.38 |
| $s = 0.75$ | 50 | 3.22 ± 0.16 | 8.80 ± 0.18 |
| $s = 0.9$ | 20 | 4.21 ± 0.30 | 8.24 ± 0.18 |
| $s = 1$*** | 1 | 3.69 ± 0.03 | 90.6 ± 0.04 |

$\texttt{Air}_{\texttt{Multi}}$ $(10^{-3})$

| Model | NFEs | $(\text{MSE}_{\boldsymbol{\mu}_y})_a$ | $(\text{MSE}_{\boldsymbol{\sigma}_y})_a$ |
|---|---|---|---|
| **Liu and Thuerey (2024)** | | | |
| DDPM T200C* | 200 | 1.76 ± 0.23 | **0.89 ± 0.12** |
| Heteroscedastic# | 1 | **1.50 ± 0.12** | 1.55 ± 0.03 |
| **DDPM** | | | |
| T100 | 100 | **2.34 ± 1.11** | **0.57 ± 0.03** |
| **EDM** | | | |
| Euler - Determ. | 40 | 1.25 ± 0.41 | 0.89 ± 0.22 |
| Stoch. | 40 | 1.13 ± 0.39 | **0.67 ± 0.11** |
| Heun - Determ. | 80 | 1.28 ± 0.42 | 0.90 ± 0.23 |
| Stoch. | 80 | **1.10 ± 0.38** | 0.71 ± 0.13 |
| **DDIM** | | | |
| T100 | 10 | 2.33 ± 1.06 | 0.83 ± 0.16 |
| | 25 | 2.16 ± 1.01 | 0.67 ± 0.07 |
| | 50 | **2.12 ± 0.99** | **0.64 ± 0.05** |
| **IR (Ours)** | | | |
| T100, $s = 0.6$, $\gamma_5$ | 41 | **2.07 ± 0.99** | **0.59 ± 0.04** |
| **TSM (Ours)** | | | |
| T100, $s = 0.5$ | 50 | 2.58 ± 1.32 | 0.79 ± 0.16 |
| $s = 0.75$ | 25 | **1.66 ± 0.53** | **0.64 ± 0.04** |
| $s = 0.9$ | 10 | 1.91 ± 0.71 | 0.70 ± 0.06 |

## E.2 QUALITATIVE RESULTS

We provide sample results for all test cases in this section. Results for all flow fields of the $\texttt{Fturb}$ study are presented in Fig. 11, while results for the $\texttt{Tra}$ test case are presented in Fig. 12. Finally, sample results for $\texttt{Air}_{\texttt{One}}$ and $\texttt{Air}_{\texttt{Multi}}$ test cases are presented in Fig. 13 and Fig. 14, respectively.

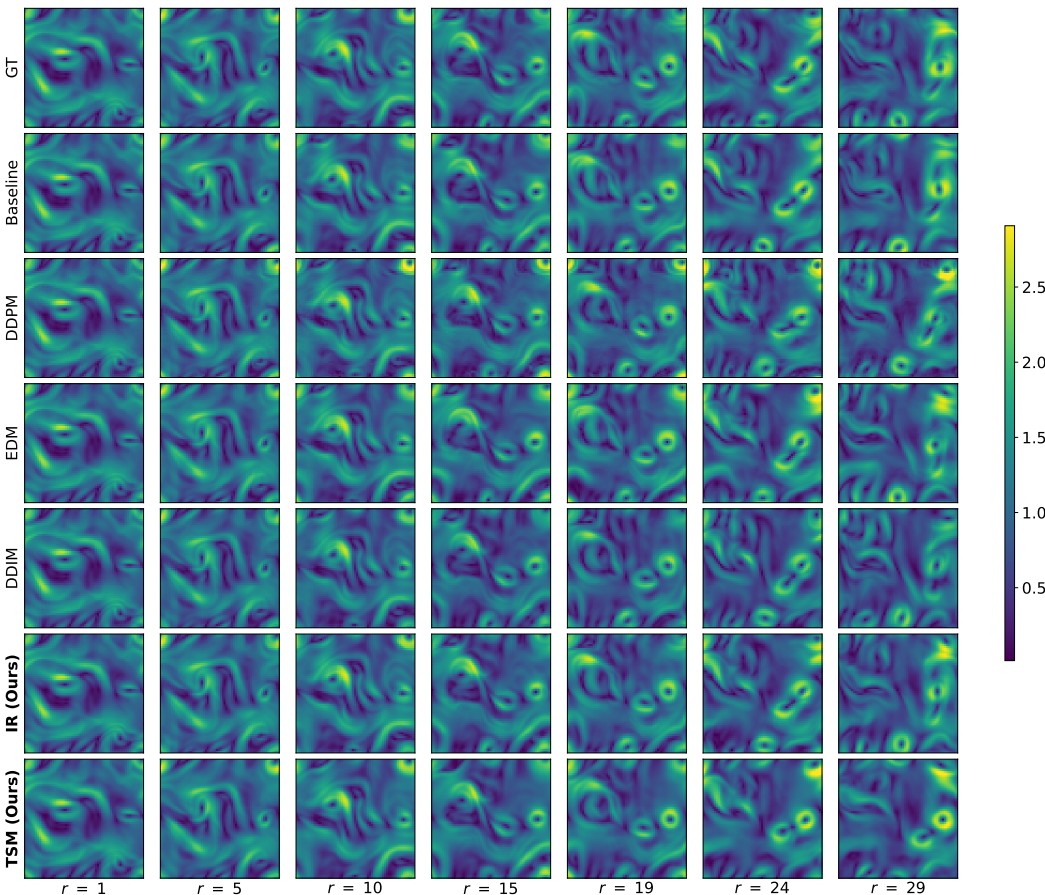

Figure 11: Sample predictions of $|u|$ from the $\mathtt{Fturb_{ext}}$ test case with $Re = 5000$. Results are based on the top performing models presented in Table 1 (right).

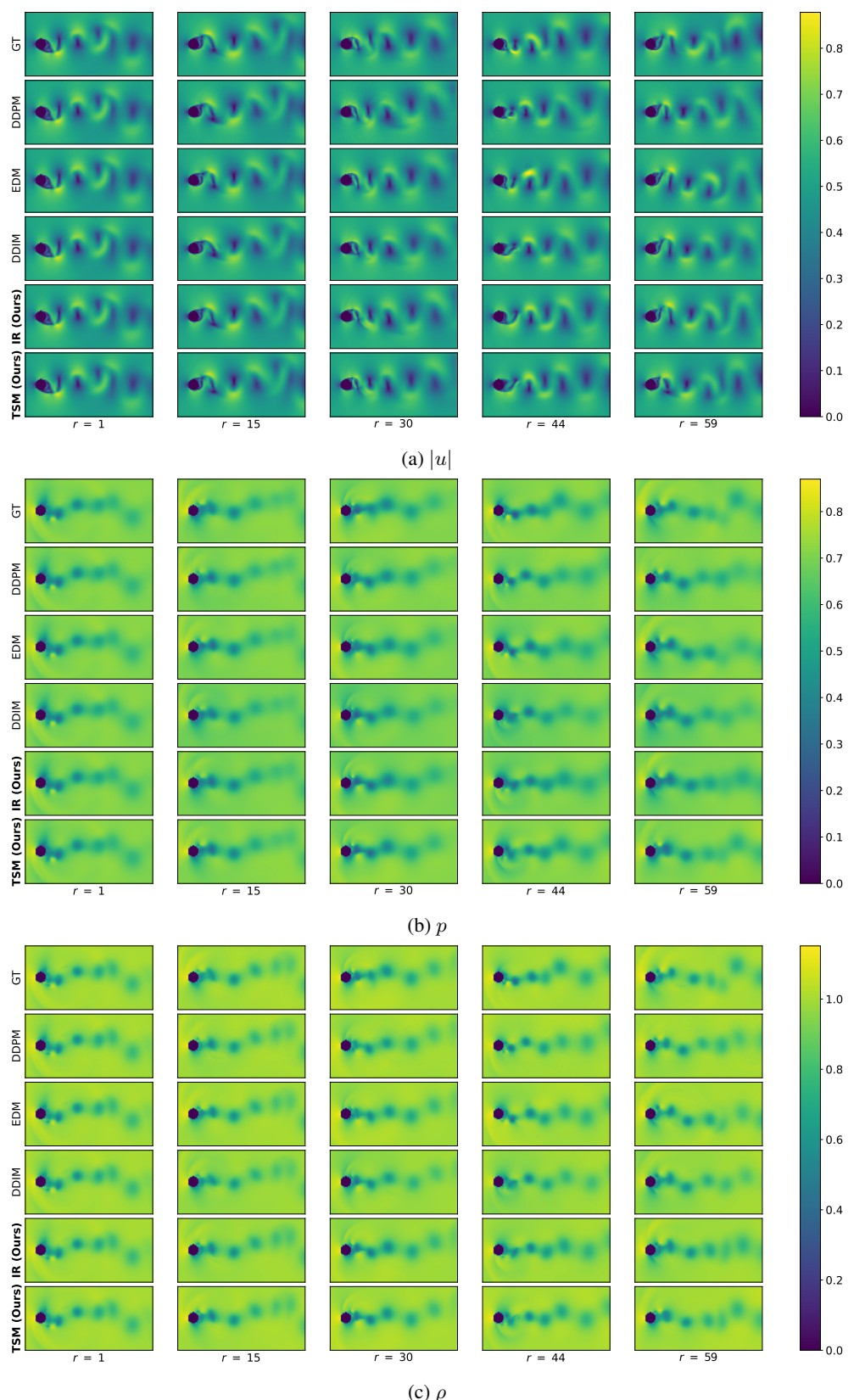

Figure 12: Sample predictions of the different flow fields from the $\mathtt{Tra_{ext}}$ test case with $Ma = 0.52$. Results are based on the top performing models presented in Table 1 (left).

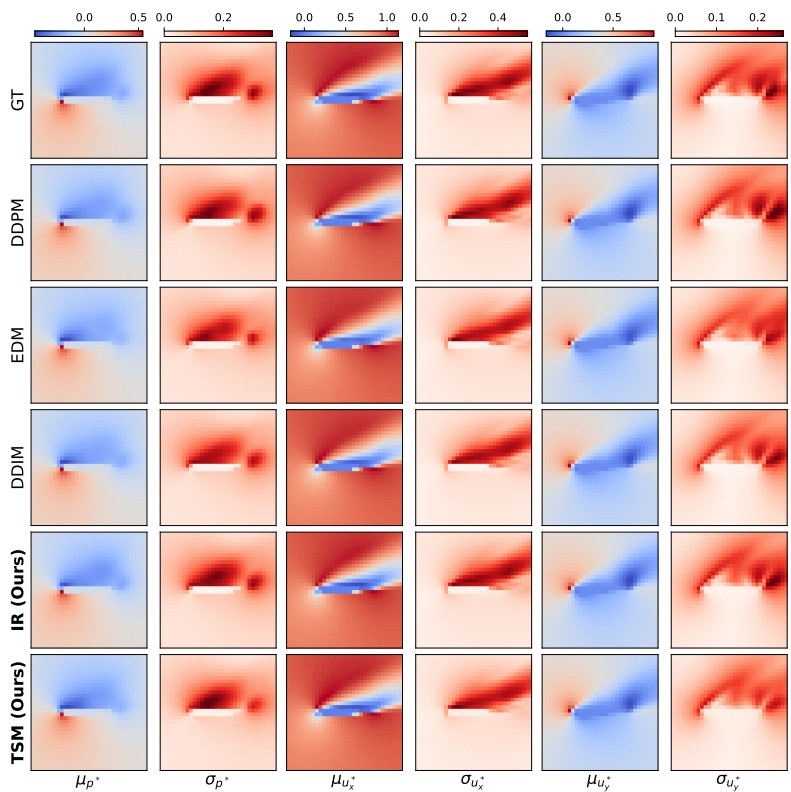

Figure 13: Sample predictions for the expectation $\mu$ and standard deviation $\sigma$ of all flow fields for the $\texttt{Air}_\texttt{One}$ case with $Re = 4.5\times10^6$. Results are based on the top performing models in Table 2 (left).

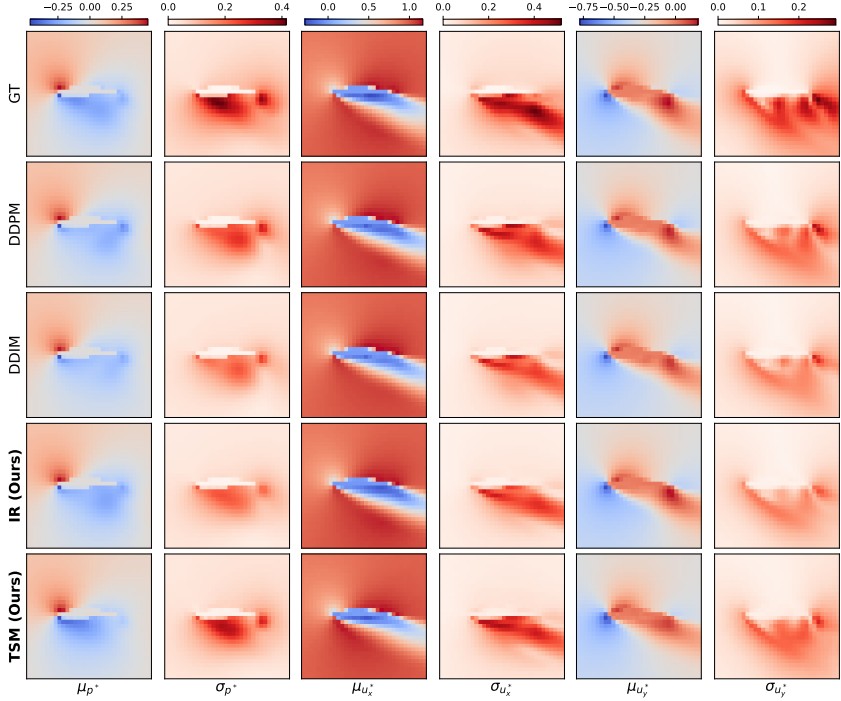

Figure 14: Sample predictions for the expectation $\mu$ and standard deviation $\sigma$ of all flow fields from the $\texttt{Air}_\texttt{Multi}$ test case for the *fx75141* airfoil with -18.94° angle of attack and $Re = 10.303\times10^6$. Results are based on the top performing models presented in Table 2 (right).

## F    DATASET OVERLAPPING FOR TRANSIENT CASES

In time-dependent problems, the proposed formulation requires training the DDPM to predict $\mathcal{T} + 1$ different steps with prediction stride $j \in [0, \mathcal{T}]$. For benchmark datasets (e.g., `Tra` Kohl et al. (2024)) that are sufficient to train for predicting next timesteps only, our formulation leads to overfitting and sub-optimal results due to the limited data, especially when the dataset is iterated as demonstrated in Figure 15a. This naïve iteration over the dataset frames means that we require a dataset that is $\times \mathcal{T}$ bigger to train without overfitting. The easiest approach to increase the dataset without the need for generating more training data is to implement overlapping, as depicted in Figure 15b. Although the demonstrated overlapping leads to more iterations per epoch depending on the degree of overlap, we found in our tests that training could be terminated much earlier, leading to almost a constant total number of training iterations.

In the `Fturb` case, enough data was present to not require significant overlapping; however, we apply max overlap (i.e., overlapping 10 frames, since $\mathcal{T} = 10$) for the `Tra` test case to circumvent overfitting issues and provide accurate results. As discussed, we usually terminated the training procedure much earlier than the number of epochs reported in Table 4 as soon as the validation loss reaches a plateau.

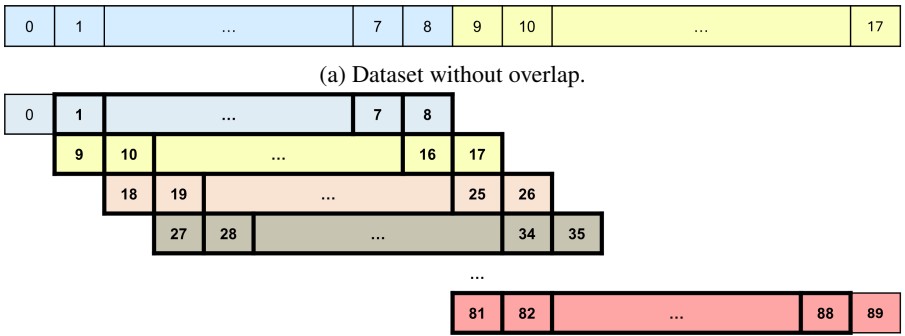

Figure 15: Illustration of dataset overlapping to circumvent limited data for training. Numbers inside cells indicate the index of the frame in the dataset, whereas the colored cells indicate one "simulation" based on the horizon $\mathcal{T}$. Each column represents a single frame but with different assigned indices for dataset iteration. The figure shows for a horizon of 8 steps (i.e., $j \in [0, \mathcal{T}]$ with $\mathcal{T} = 8$) how max overlapping (max overlap $= \mathcal{T} = 8$ frames) can lead to increase in the considered dataset from 18 to 90 frames each with a unique label within a "simulation" (i.e., color). Frames with a bold border are provided more than one label.

## G    HYPERPARAMETER TUNING

Our methods rely on hyperparameters that play a crucial role in determining their performance in terms of accuracy and speed. Unlike DDIMs, which provide a trade-off between speed and accuracy, our methods are tuned using a heuristics-based approach to efficiently identify near-optimal hyperparameter configurations with minimal evaluations. This approach depends on the specific case under consideration, whether it is a spatio-temporal deterministic dataset or a stochastic one with a complex distribution. The hyperparameters can be determined conveniently based on the following considerations, and are kept constant for each case during inference runs thereafter.

**Deterministic test cases**

- **TSM:** The search for the optimum $s$ value typically begins within the range $[0.5, 1]$, especially if a large number of diffusion steps $T$ is chosen, to attain higher speedup. Figure 3 provides several insights for the optimal combination of $s$ and $T$ for both transient and steady-state cases. First, we evaluate both extremes of $s$ within the specified range and then employ a standard line search approach to arrive at an optimal value of $s$, requiring at most

two or three additional evaluations. Further, we also give priority for models with low $T$ to minimize the NFEs required for inference.

- **IR:** we usually start with $x_{init} \sim \mathcal{N}(0, I)$ and run our greedy optimization algorithm (Algorithm 3) with low $N$ to obtain an efficient $\gamma$ with as low NFEs as possible. As demonstrated in Figure 4b, $N = 5$ often yields highly accurate results for our transient cases. Then, $N$ can be gradually increased to explore longer schedules that has the potential to enhance the overall accuracy with little increase in NFEs.

**Stochastic test cases**

- **TSM:** We follow the same procedure as before; however, we refrain from extreme $s$ values as they often lead to low-fidelity, noisy outputs (see Table 9). Therefore, the search region for $s$ is restricted to $[0.5, 0.9]$, with the lower bound further reduced for smaller $T$ values.

- **IR:** From early tests, it is found that $x_{init}$ is optimal when obtained through truncated sampling (see Algorithm 1) of a pre-trained DDPM with $s \geq 0.5$. While this approach typically produces highly accurate initial outputs, the results often exhibit noticeable noise. Consequently, a refinement schedule $\gamma$ with $N < 5$ steps is utilized to eliminate the noise, while simultaneously improving or preserving the accuracy of the noisy initial input.

Empirical results across all datasets, as presented in Tables 7, 8, and 9, demonstrate that our methods require relatively few evaluations to arrive at hyperparameter configurations with highly competitive results. Compared to competitive EDMs, the reduced number of associated hyperparameters in addition to our heuristic approach provide our methods with a distinct competitive advantage, alongside the other advantages outlined previously.

