# OpenReview forum: "Improved Sampling Of Diffusion Models In Fluid Dynamics With Tweedie's Formula"
_ICLR.cc/2025/Conference — ICLR 2025 Poster_

### Official Review · Reviewer_6cX6 · 2024-10-28

**Soundness:** 2
**Presentation:** 3
**Contribution:** 2
**Rating:** 6
**Confidence:** 3

**Summary:**

The paper addresses the high computational requirements of existing diffusion-based techniques for modelling dynamical systems. They propose two sampling techniques that lead to good sample quality with only a few NFEs. The first one requires modifications to the training process and is performed by truncating the diffusion process close to the clean data. The second one is compatible with pre-trained DDPMs and proposes an iterative refinement based on Tweedie’s formula. The paper provides extensive experimental evidence on three datasets: incompressible and compressible turbulent flow (2D) and airfoil flow simulation (3D).

**Strengths:**

1. **Relevant topic.** Lately, there has been increasing interest in modelling dynamical systems with diffusion models due to their probabilistic nature. However, many works (Kohl et al. [1], Shysheya et al. [2], Lippe et al. [3]) acknowledge that diffusion models tend to be computationally costly, and techniques that reduce this cost would be greatly beneficial. This is exactly the problem this paper aims to address.
2. **Clear distinction from other works.** The paper clearly delineates its contributions from the already existing work, and how the proposed sampling techniques differ from other approaches.
3. **Good experimental evidence.** The paper provides good empirical evidence, with experiments on three diverse datasets, and using a wide range of metrics.
4. **Well-structured, clear writing.** Overall, I found the structure and writing clear.

[1] Kohl, G., Chen, L., & Thuerey, N. (2023). Benchmarking Autoregressive Conditional Diffusion Models for Turbulent Flow Simulation.

[2] Shysheya, A., Diaconu, C., Bergamin, F., Perdikaris, P., Hern'andez-Lobato, J.M., Turner, R.E., & Mathieu, E. (2024). On conditional diffusion models for PDE simulations.

[3] Lippe, P., Veeling, B.S., Perdikaris, P., Turner, R.E., & Brandstetter, J. (2023). PDE-Refiner: Achieving Accurate Long Rollouts with Neural PDE Solvers. ArXiv, abs/2308.05732.

**Weaknesses:**

1. **Unclear how the method generalises to other settings.** I acknowledge that the aim of the paper is to study the efficacy of the proposed sampling methods in the context of dynamical systems. As the authors mention, the fact that they lead to good results is probably due to:
- The characteristics of the data distribution—states with fairly coarse resolutions, where the high frequency information has been lost during downsampling.
- The task considered—predicting next step (forecasting) distribution, which is predominantly unimodal.

  I’d be curious to know whether the sampling methods remain applicable
  - for other tasks that might require modelling multimodal distributions: e.g. some sort of inverse problem (reconstruct field based on sparse, partial observations, compatible with multiple solutions), which is also a task of interest in dynamical system modelling,
  - or for other datasets: potentially still PDEs but not as coarsened and with more complicated nonlinear behaviour, or images.

  One could argue that Air_multi requires sampling from a more complicated distribution, and there the improvements are not as pronounced as in other settings. Perhaps you could include a brief discussion on this, or a preliminary experiment on a more complex PDE dataset, such as Kuramoto-Sivashinsky (see **Q1**)?
2. **Counter-intuitive comparison to DDPM.** Maybe this is not necessarily a weakness, but I find it counter-intuitive that these sampling methods outperform DDPM, given that DDPM (or at least the continuous-time frame formulation) has stronger theoretical foundations (Similarly to how DDIM is posed as a framework where you can trade off computation for sample quality, not gain in both). I would have expected the same accuracy with significantly fewer NFEs to be possible, but not both. Is it possible that the DDPM model could be further tuned to achieve similar performance and maybe the DDPM baselines are not optimal? Maybe this is where a comparison with EDM would have been beneficial, as it provides a more principled approach of setting up diffusion models.
3. **Lack of theoretical guarantees.** While the methods seem effective in practice, the paper does not provide any theoretical guarantees. I realise this might be hard to derive, but this makes the applicability of the methods more ad-hoc. This also reflects in the empirical investigation, where there is no clear recipe for what works best and how one should choose the hyperparameters optimally.
Also when making statements such as “However, we assert that
our methods will consistently improve over ancestral sampling, as demonstrated in our experiments.” - This was only shown in three experiments, and the paper does not contain guarantees that this would always hold, so I would avoid over-generalising.
4. **Lack of analysis of the stability of longer rollouts.** The paper provides several metrics to analyse the performance of the sampling methods, but none provides intuition/results about how the metrics evolve in time for the transient datasets (Tra and Fturb) (e.g. per-time-step MSE or per-time-step correlation). In particular, I’d be interested to compare the performance of the benchmarks to your methods on rollouts longer than what the model has been trained on. (This could, for instance, be tested on Tra_long from Kohl et al. [1]).

    I think that one potential weakness of these sampling schemes is that they lose more of the high frequency information than the traditional sampling schemes. Maybe this doesn’t affect the short rollouts significantly but it might negatively impact the accuracy of longer term rollouts (see Lippe et al. [3]). It would also be good to check that if extended significantly beyond the training range, the proposed sampling methods still generate physically plausible states, and outperform the baselines.
5. **Lack of a comparison to EDM.** While I agree that a comparison to, for example, distillation techniques is outside the scope of the paper, I think EDM is relevant as a baseline. The fact that it is “designed to handle more complex stochastic data with multimodal distribution” does not mean it is not relevant for cases that do not exhibit much stochasticity. And it shouldn’t incur a different computational cost at training time. I think a comparison to EDM would be very useful to the community to figure out what the fastest and most accurate way to set up diffusion models for dynamical systems is. If the methods proposed here outperform EDM, this makes the paper stronger. If they don’t, I think this would also be a valuable result, potentially implying that EDM is a robust technique (regardless of data distribution) that should be used as a “first thing to try” as opposed to spending time and resources on hyperparameter tuning of more ad-hoc techniques.
   Maybe the authors could include a comparison to EDM on one or two experiments?
6. **Unclear experimental setup.** I found it hard to figure out some experimental details in certain places. I am aware these details exist in other papers (Kohl et al. [1]), but for ease of interpreting the results, it would help to include some more details, potentially as a brief table in the appendix which summarises the most important dataset characteristics. For example:
   - Tra - What is the Mach number range of the training trajectories? Is it also Ma $\in [0.53, 0.63] \cup [0.69, 0.90]$ as in Kohl et al.?
   - Tra - How many trajectories are there per Mach number?
   - Fturb - I am slightly confused about the number of states within each trajectory for this dataset. You mention that each simulation contains 51 temporal states, but do you just consider 30 out of these for the test results? And what do you mean by “AR sampling is employed for $R = 30$ timesteps … for two sequences per $Re$?”
   - Fturb - you are feeding in the $Re$ number as conditioning information to the model as in Kohl et al., right?

**Minor**

7. **Comparison to DDIM (L306)** - You say that “IR should supersede the deterministic DDIM sampling regarding accuracy and NFEs” due to its stochastic nature. While I agree that stochastic sampling is beneficial because it can correct previous errors in sampling, Karras et al. [4] mention that, in practice, the situation is more complex because approximating the extra Langevin term introduces error in itself (see Section 4 Stochastic sampling). Thus, I would not say there is any guarantee that IR would supersede DDIM in all scenarios.
8. **Lack of discussion on $j$.** I find it interesting that the optimum $j$ value varied so much between methods, as shown in Tables 6 and 7, but you do not include any discussion about this. Do you have any insight about why this might be the case?
9. **Small typos**, such as L291 “to be evaluated”, L776 “presnted”, L772 “where” missing at the beginning of the line, L819 “allow” rather than “allows”, L840 “are not needed” etc.

Overall, I think the paper is clearly written and structured, and generally presents convincing empirical evidence. However, I think its quality could be significantly improved by including experiments on longer rollouts, a comparison to EDM, and potentially clarifying the regimes in which the techniques are effective (with the inclusion of negative results if necessary).

[4] Karras, T., Aittala, M., Aila, T., & Laine, S. (2022). Elucidating the Design Space of Diffusion-Based Generative Models. ArXiv, abs/2206.00364.

**Questions:**

1. **Regarding W1** - I’d be curious to see how the methods perform on dynamical systems with more complicated nonlinear dynamics, for example the Kuramoto-Sivashinsky equation used in PDE-Refiner (Lippe et al. [3]). That is a fourth-order non-linear equation where correctly capturing the high frequencies seems to be more important than in other settings, so it would be interesting to see how you compare there with benchmarks such as PDE-Refiner [3] (which can also be interpreted as a refinement of an MSE-trained one-step prediction).
2. **Regarding W2** - Do you have any intuition why these methods outperform DDPM and whether the DDPM baseline could be improved?
3. **Regarding W4** - Could you provide some experiments that test the long rollout performance of these sampling methods vs. traditional ones? Including frequency spectra of generated states would also help.
4. **Regarding Minor 2** - It seems that in general the optimum $j$ for these methods is between [2, 4], but have you noticed any significant patterns? Were there differences between interpolation and extrapolation tasks?
5. **Eq (5).** I am not sure I understand this equation. I will omit the $\theta$ in the $p_{\theta}$ subscript because the equations do not render correctly. If $p^T(x_T, \mathbf{x}_0, j) = \mathbf{x}(j \cdot \delta t)$, then wouldn’t $\mathbf{x}(2j \cdot \delta t) = p^T(x_T, \mathbf{x}(j \cdot \delta t), j) = p^T(x_T, p^T(x_T, \mathbf{x}_0, j), j)$ (i.e., we still start from white noise $x_T$, but we now condition on the output of the previous step)? In my mind $\mathbf{x}(\tau_f) = p^T(x_T, p^T(...p^T(x_T, \mathbf{x}_0, j)...), j)$, but maybe I didn’t interpret the equation correctly.
6. When comparing the results in Table 5 vs. Tables 6 and 7 - Shouldn’t the metrics corresponding to, for example, Fturb TSM T80 s =0.75 ($j=2$) from Table 5 ($3.63 \pm 1.95$) be the same as in TSM T80, s=0.75 with optimal $j=2$ ($3.43 \pm 3.03$) Table 7? What’s the difference between these settings?

**----Update after rebuttal-----**

The majority of my concerns have been addressed and I believe the paper offers a useful empirical investigation into how to speed up sampling with diffusion models for physics-based simulations. As such, I increased my score to 6.

---

> ### Author Response · Authors · 2024-11-21
> **Part (1/4)**
>
> We thank the reviewer for their thorough review and the invaluable feedback and suggestions. We address their comments and questions below.
>
> **W1 & Q1 - Generalization to other settings :** We thank the reviewer for their insights and suggestions. Regarding $Air_{multi}$, we see that our models at least maintain the accuracy of the baseline DDPM and significantly reduce NFEs from 100 down to 41 and 10 for IR and TSM, respectively. Thus, we believe that our proposed approaches are promising for application to datasets with complex distributions.
> Furthermore, we agree that evaluating our approaches on systems with more complex nonlinear dynamics, such as the Kuramoto-Sivashinsky (KS) equation, would provide valuable insights, especially for capturing high-frequency components. While our work does not directly include KS, we address similar challenges in the $Tra$ case, where shock waves introduce sharp, localized discontinuities that correspond to high-frequency energy in the spectral domain, and the $Fturb$ case, which "exhibits extreme events in the form of bursts in kinetic energy and dissipation" [1]. The success of our methods in capturing these features suggests their potential applicability to similarly complex PDEs like KS, yet we still believe it would be interesting to evaluate the performance gains on this dataset and compare against benchmark results.
> However, extending our approaches to KS or similar systems would require careful optimization of DDPM, IR, and TSM parameters, which is beyond the scope of this rebuttal. However, we appreciate the suggestion and would be happy to explore this in the final version.
>
> [1] Clustering-Based Identification of Precursors of Extreme Events in Chaotic Systems. In International Conference on Computational Science (pp. 313-327). Cham: Springer Nature Switzerland. (2023)
>
> **W2 & Q2 - Counter-intuitive comparison to DDPM.** While we agree with the reviewer that our improvements in terms of both speed and accuracy in some experiments might be surprising given the strong theoretical foundations for general DMs, our improvements arise from domain-specific adaptations. For this reason, we chose our experiments to exhibit different levels of difficulty and complex dynamics in both steady-state and transient scenarios to provide adequate empirical evidence for the efficacy of our approaches.
> Furthermore, we believe that our baseline DDPMs are competitive and well-tuned for our datasets. In fact, in our methodology, we first optimize the best possible baseline DDPM in comparison to (benchmark) deterministic baselines, then we tune for $s$ in TSMs and \{$\gamma$, $x_{init}$\} for IR sampling. Therefore, we are confident that our DDPM results are fair and representative. Additionally, we include a comparison against EDMs in our response below to **W5**.
>
> **W3 - Lack of theoretical guarantees.** We agree that deriving theoretical guarantees is challenging. Further, we provide practical heuristics that guide hyperparameter selection effectively across datasets for IR and TSM.
>
> - Deterministic test cases
>     - IR: we usually start with $x_{init} \sim \mathcal{N}(0,I)$ and run our greedy optimization algorithm with low N (with $N = |\gamma|$) to obtain an efficient $\gamma$ with low NFEs. As demonstrated in Figure 4(b), $N=5$ often yield very good results for transient cases. $N$ can then be gradually increased to explore other schedules that might improve the accuracy with little increase in NFEs.
>     - TSM: For higher speedup, the search for the optimum $s$ value typically begins within the range $[0.5, 1]$, especially if a large number of diffusion steps $T$ is chosen. We believe that Fig (3) provides several insights for the optimal combination of $s$ and $T$. We first test for both extremes of $s$ and then follow a standard line search approach to arrive at an optimum value for $s$, requiring additional 2 or 3 evaluations at most. We also give priority for models with low $T$ to minimize the NFEs required for inference.
> - Stochastic test cases
>     - IR: $x_{init}$ is optimal when obtained through truncated sampling of a pre-trained DDPM with $s >= 0.5$. The output from this approach is usually highly accurate but exhibits clear noise; thus it usually takes $N < 5$ for $\gamma$ to arrive at a clear output while improving or retaining the accuracy of the noisy input.
>     - TSM: We follow the same procedure as before; however, we refrain from extreme $s$ values. Therefore, our search is restricted to $s \in [0.5, 0.9]$, or smaller lower bound for low $T$ values.
>
> We will dedicate an additional section in our main text for an elaborate approach to optimizing the hyperparameters related to our proposed approaches. Moreover, we acknowledge that our experiments don't guarantee universal applicability; therefore, we will revise the generalizing statement to reflect the scope of our findings in the updated version of our paper.

---

> ### Author Response · Authors · 2024-11-21
> **Part (2/4)**
>
> **W4 & Q3.** We thank the reviewer for their suggestion regarding the stability for longer rollouts of our approaches. In addition to time-averaged metrics, in Fig. 7 (Appendix C), we show how each of the top-performing models from Table 2 correlate with the ground truth solution trajectory over time by calculating the Pearson correlation coefficient for the absolute velocity $\rho (|u|)$ at each timestep $r$. Also, in Fig. 2 (b), we report $\rho$ for the time evolution of the domain-wide kinetic energy, which describes the total kinetic energy of the system.
> Additionally, as requested, we include more analysis with regards to the temporal stability of our methods compared to traditional ones. We estimate the temporal stability for both $Tra_{long}$ (as defined in [2]) and $Fturb_{long}$ (combines both $ext$ and $int$ regions but with $R = 120$ instead of $R = 30$, more details will be included in an additional appendix for detailing all datasets) by calculating the rate of change of each flow field $x$ using $||(x_r - x_{r-1})/\Delta \tau||$ [2] and comparing against the ground truth simulation (GT). In summary, for $Tra_{long}$, our stability parameter remains bounded within the range $[0.015, 0.02]$ for both the ground truth (GT) and all models. While DDIM, DDPM, and IR slightly exceed the lower bound, they maintain overall stability. For $Fturb_{long}$, the stability parameter for the GT and all models is constrained within $[0.1, 0.15]$, with the exception of a minor deviation observed in DDPM. *The corresponding Figure will be included in our revised paper*. Contrary to $\rho (|u|)$, this parameter is useful for long rollouts as it helps identify whether a solution trajectory remains physical even after no longer being correlated to the GT. These additional results show that our proposed methods exhibit temporal stability for simulation trajectories significantly longer than the trajectories used for training.
>
> [2]Benchmarking Autoregressive Conditional Diffusion Models for Turbulent Flow Simulation. In: ICML (2024).
>
> **W5 - Lack of a comparison to EDM.** As requested, we trained EDMs for three out of our four datasets, namely $Tra$, $Fturb$, and $Air_{One}$, and compare their performance against the top-performing models reported in Tables 1 and 2 (left). We use the same training hyperparameters (see Table 3) and network architecture as in Table 4. Our implementation of EDMs is based on the work by Karras et al. (2022). We implement their Algorithm 1 (deterministic sampler) and Algorithm 2 (Stochastic sampler) using 1st-order Euler and 2nd-order Heun's methods with design choices from the last column of Table 1 (Karras et al., 2022). We also consider preconditioning and a weighted loss function using the parameters recommended by the authors. For each case, we ran the model using different combination of deterministic/stochastic sampler and Euler's/Heun's method. For the stochastic sampler, the parameters $\{S_{churn},S_{tmin}, S_{tmax}, S_{noise} \}$ were non-comprehensively tuned to attain the best possible results. TA-MSE vs NFEs ($\in [1,200]$) curves for these various samplers will be included in our revised manuscript.
> In the tables below, we provide the updated Tables 1 and 2 (left) with the additional best EDM model. We summarize these finds:
>    - $Tra$ - EDM outperforms all probabilistic and deterministic models with 4 NFEs only.TSM achieves nearly comparable accuracy while remaining the sole model capable of single-step inference. Given the negligible accuracy difference from the best-performing model, we hypothesize that a more accurately pre-trained DDPM could enable IR to achieve more competitive results with low NFEs.
>    - $Fturb$ - EDM outperforms DDPM and DDIM in both accuracy and NFEs but achieves comparable performance to TSM, despite being $10\times$ slower. IR, however, remains the most accurate model for this dataset.
>    - $Air_{One}$ - EDM performs poorly in comparison to all other models. We believe the reason for this suboptimal performance is that the hyperparameters (including loss function weighting, scaling, and diffusion-related parameters) might not be optimal for this case and thus would require comprehensive tuning for improved results. Indeed, we believe that these author-recommended settings for EDMs are not expected to perform well in all applications (e.g., see [3]).
>    - In conclusion, EDMs are a great alternative to standard DDPMs, yet they don't provide significant improvements across different fluid dynamics problems compared to other baselines and our proposed approaches. We believe an interesting avenue for future research would be the combination of our proposed methods with EDMs to potentially enhance their performance and most importantly facilitate single-step inference.
>
> We report these new findings in our updated manuscript.
>
> [3]Scaling Rectified Flow Transformers for High-Resolution Image Synthesis. In: ICML (2024).

---

> > ### Author Response · Authors · 2024-11-21
> > **Part (3/4)**
> >
> > | **Model**                              | **NFEs** | **Tra ext ($10^{-3}$)** | **Tra int ($10^{-3}$)** | **Model**                              | **NFEs** | **Fturb ext ($10^{-2}$)** | **Fturb int ($10^{-2}$)** |
> > |----------------------------------------|---------|-------------------------|-------------------------|----------------------------------------|---------|---------------------------|---------------------------|
> > | ACDM T20 (Kohl et al., 2024)           | 20      | 2.3 ± 1.4               | 2.7 ± 2.1               |                                        |         |                           |                           |
> > | $UNet_{ut}$ (Kohl et al., 2024)        | 1       | 1.6 ± 0.7               | 1.5 ± 1.5           | Baseline                               | 1       | 3.95 ± 3.84               | 4.82 ± 5.23               |
> > | DDPM T100                              | 100     | 3.0 ± 2.7               | 4.1 ± 3.7               | DDPM T80                              | 80      | 6.16 ± 6.54               | 5.27 ± 5.55               |
> > | **EDM**                            |       4  |    **1.3 $\pm$ 1.3**                     |     **1.1 $\pm$ 1.0**                   | **EDM**                           | 10      | 4.75 ± 4.53               | 4.78 ± 4.26               |
> > | DDIM T20                               | 10      | 3.2 ± 2.7               | 4.2 ± 3.9               | DDIM T80                              | 40      | 4.31 ± 4.62               | 6.97 ± 6.84               |
> > | IR T100 - $\mathcal{N}$ $\gamma_1$ (Ours) | 5     | 1.6 ± 1.3               | 2.0 ± 1.7               | IR T80 - $\mathcal{N}$ (Ours)         | 10      | **2.93 ± 3.34**           | **1.70 ± 1.63**           |
> > | TSM T100 $s=1$ (Ours)                  | 1       | **1.2 ± 1.1**           | 1.5 ± 1.5           | TSM T100 $s=1$ (Ours)                 | 1       | 5.00 ± 5.52               | 4.39 ± 4.76               |
> >
> > For the $Air_{One}$ case:
> >
> > | **Model**                              | **NFEs** | **$(MSE_{{\mu},y})_{a}$ ($10^{-4}$)** | **$(MSE_{{\sigma},y})_{a}$ ($10^{-4}$)** |
> > |----------------------------------------|---------|------------------------------------------------------|------------------------------------------------------|
> > | DDPM T200C$^\ast$ (Liu and Thuerey, 2024) | 200     | 3.79 ± 0.27                                          | 8.40 ± 0.69                                          |
> > | DDPM T200                              | 200     | **2.88 ± 0.26**                                      | 7.05 ± 0.22                                          |
> > | **EDM**                            | 20      | 8.13 ± 1.28                                          | 10.1 ± 1.67                                          |
> > | DDIM T200                              | 100     | 3.68 ± 0.44                                          | 7.24 ± 0.25                                          |
> > | IR T100 - $s=0.6$ $\gamma_\text{5}$ (Ours) | 41 | **2.87 ± 0.32**                                      | 6.76 ± 0.20                                          |
> > | TSM T100 $s=0.9$ (Ours)                | 10      | 3.30 ± 0.39                                          | **5.89 ± 0.33**                                      |
> >
> >
> > **Unclear experimental setup.** We agree with the reviewer that a more concise presentation of the experimental setup would improve clarity. We will dedicate an additional appendix to summarize the primary parameters for the different datasets for the sake of completeness.
> > Regarding the examples mentioned:
> > - **Tra**: Yes, we use the exact same training trajectories as in the benchmark paper.
> > - **Tra**: For each $Ma$, there is a single trajectory with $R = 500$. However, each is split into 2 consecutive trajectories with $R = 60$, starting from $r = 250$ for the test regions *ext* and *int*.
> > - **Fturb**: For each $Re$, there are 240 trajectories (each starting from a different initial state). Each trajectory has 51 states (i.e., $R = 51$). However, during testing, we consider two trajectories of 30 states each for testing as we observed during early experiments that the predictions quickly de-correlate from the ground truth for longer rollouts.
> > - **Fturb**: True. $Re$ is a scalar condition provided as an input to the network as a 2D constant field, similar to how we handle $Ma$ for the *Tra* case.
> >
> > **Minor 1 - Comparison to DDIM.** We indeed agree that the dynamics of stochastic sampling is complicated in practice and that our statement might not reflect our intended meaning. Our statement is intended to reflect that IR should supersede DDIM based on the empirical observations from our experiments, which are focused on select fluid dynamics problems. Since the benefits of stochastic sampling are case-dependent, we will update our statement to reflect that trade-off between stochastic and deterministic approaches may vary across domains and thus doesn't guarantee that IR will always outperform DDIM.

---

> > > ### Author Response · Authors · 2024-11-21
> > > **Part (4/4)**
> > >
> > > **Minor 2 & Q4 - Lack of discussion on $j$.** The optimum $j$ value is an interesting finding in our results. While we don't see a clear pattern between the different models and experiments, we observe that the most accurate models for $Tra$ has $j_{optimal}\leq4$ (with few exceptions) and $Fturb$ has $j_{optimal} \in \{2, 6\}$. This variability suggests that the optimal $j$ depends not only on the specific surrogate model but also on factors such as timestep size $\delta \tau$, trajectory length $R$, and the inherent complexity of the dataset.
> > > Our analysis remains focused on identifying optimal $j$ values for specific models and experiments. Importantly, these findings demonstrate that next-step sampling (i.e., $j=1$) is not always the most accurate choice, and larger strides can achieve competitive accuracy with enhanced parallelization. While we have not conducted a fully comprehensive study of $j_{optimal}$, we hope this work motivates further research into stride conditioning for faster and more accurate inference in spatio-temporal CFD problems.
> > >
> > > **Minor 3 - typos.** We thank the reviewer for pointing out these typos. We will carefully proofread the manuscript to address the noted errors.
> > >
> > > **Q5:** We thank the reviewer for pointing out this typo in Eq. 5. Indeed, the first argument for $p^T$ should always be $x_T$ while the second argument depends on the output of the previous DDPM iterative sampling function $p^T$.
> > >
> > > **Q6:** There are no differences between the settings in Table 5 and Tables 6 and 7. In Table 5, the TA-MSE values are averaged over both $ext$ and $int$ regions (as outlined in the caption of Table 5), while the results in Table 6 and 7, the TA-MSE values have been reported separately for each region.

---

> > > > ### Comment · Reviewer_6cX6 · 2024-11-22
> > > >
> > > > Thank you for your clarifications! I’m looking forward to reviewing the revised manuscript. Below are a few additional comments in response to your replies:
> > > >
> > > > **W1 & Q1**: Understood. If time permits, exploring this in the final version would be very useful.
> > > >
> > > > **W2 & Q2**: Agreed—it makes sense, and I think you made the right decision to adjust the paper’s title to emphasise the focus on physics-based simulations more clearly.
> > > >
> > > > **W3**: This is very useful, especially given the heuristic nature of the approach. Including a clear "recipe" for tuning the hyperparameters would greatly enhance the usability of the method.
> > > >
> > > > **W4 & Q3**: Great, I think such a figure would be beneficial to showcase that the predictions remain physical. It would be even better if the figure plotted results over trajectory length or time steps, similar to Fig. 10 in Kohl et al.
> > > >
> > > > **W5**: This is interesting. While additional tuning might make the EDM results more consistent even for the more challenging datasets, I understand that this is beyond the scope of the current work. Still, the fact that your method achieves better performance with less complex fine-tuning is a valuable part of the contribution.
> > > >
> > > > **W6**: Regarding Fturb, are the test results based solely on two test trajectories (i.e., two initial conditions)? If so, wouldn’t it make sense to use a test dataset with more initial conditions to better assess the generalisation capabilities?

---

> > > > > ### Author Response · Authors · 2024-11-25
> > > > >
> > > > > We thank the reviewer for their interest in our new results (revised manuscript is now available) and their follow up comments.
> > > > >
> > > > > > Regarding Fturb, are the test results based solely on two test trajectories (i.e., two initial conditions)? If so, wouldn’t it make sense to use a test dataset with more initial conditions to better assess the generalisation capabilities?
> > > > >
> > > > > The $Fturb_{ext}$, $Fturb_{int}$, and $Fturb_{long}$ cases use two initial conditions per $Re$ and thus have a total of 6 trajectories, but with a much longer rollout length for $Fturb_{long}$. More details can now be found in Appendix A in the revised manuscript. While we could further increase the number of initial conditions per $Re$, we found in earlier tests that using 4 initial conditions per $Re$ yields very similar results with negligible variation in performance metrics. Therefore, to reduce the computational costs, we limit our assessment to two initial conditions only, which remains a fair and representative assessment of the generalization capabilities of the models.

---

> > > > > > ### Comment · Reviewer_6cX6 · 2024-11-25
> > > > > >
> > > > > > **Number of test trajectories** Ok, I see. If the authors did not notice much difference between the results on 2 vs. 4 trajectories per $Re$, then it might be that in the case of Fturb there aren't any initial conditions that are "harder" to solve than others.
> > > > > >
> > > > > > The reason why I am asking is because in other datasets (and I am going to return to the KS example here), the literature has reported that, depending on the initial conditions, some trajectories might be easier or harder to roll out from (see **Uncertainty estimation** in PDE-Refiner). Hence, in there it is crucial to make sure that the testing is performed on varied initial conditions.
> > > > > >
> > > > > > As an extension to what you did, you could explore this with just one dataset (e.g. Fturb), and explore the uncertainties provided by the diffusion model.
> > > > > >
> > > > > > **Updated manuscript** Thank you for providing the updated manuscript, there are a couple of typos which I encourage the authors to resolve (e.g. L343 dataset comprises includes $u_x$, L704 informatioon, etc.). But otherwise, it looks good. I am looking forward to the experiments you will add if the paper gets accepted.
> > > > > >
> > > > > > The majority of my concerns have been resolved and I think the paper is a good contribution to the community. While I agree with reviewer TSeh that the contributions are tailored to the fluid dynamics community, I also note that lately there has been an increasing interest in the machine learning community to tackle fluid dynamics modelling. As such, I will increase my score to 6, because I think the paper could be useful to ML researchers looking to speed up sampling in diffusion models aimed at physics-based simulation.

---

> > > > > > > ### Author Response · Authors · 2024-11-26
> > > > > > >
> > > > > > > Thank you for your additional comments and for recognizing the contributions of our work.
> > > > > > >
> > > > > > > **Number of test trajectories:** We have not conducted a deep study on the impact of varying initial conditions in the $Fturb$ case, but we agree that it is a valuable extension to further explore the accuracy of diffusion models on rollouts with different initial conditions. We will consider incorporating this analysis in our final paper. Thank you for this insightful suggestion.
> > > > > > >
> > > > > > > **Updated manuscript:** Thank you for pointing out these typos. The manuscript has been updated accordingly.

---

### Official Review · Reviewer_fSso · 2024-10-28

**Soundness:** 3
**Presentation:** 2
**Contribution:** 2
**Rating:** 5
**Confidence:** 3

**Summary:**

This paper introduces Truncated Sampling Models (TSMs) and Iterative Refinement (IR) to improve the efficiency and fidelity of Denoising Diffusion Probabilistic Models (DDPMs) for fluid simulations by enabling reduced steps sampling through truncation of the diffusion process. These methods significantly reduce inference time and improve stability over long rollout horizons for turbulent flow and airfoil simulations.

**Strengths:**

- Interesting topics – Generative diffusion models as surrogate model for fluid simulations.
- Clear motivation – Reduce computation costs in the generative process using a novel reverse sampling approach.

**Weaknesses:**

- The paper lacks clarification of superiority of the proposed methods over other surrogate models, such as neural operators which are currently the most widely used ML-based surrogate models in fluid dynamics for speed and accuracy. Given that a primary contribution of this paper is reducing time costs, a more thorough comparison with advanced neural operators – either highlighting the proposed method’s improved accuracy at similar time costs or its time efficiency at comparable accuracy – would strengthen the argument. However, there are few comparisons with neural operators in the experiments in main text; although Unet is included, more advanced neural operators should be included. Additionally, the proposed method does not appear to clearly outperform Unet.
- The title may mislead some readers, as “physics-based simulations” implies a broad range of applications, while the paper is mostly on fluid dynamics. To improve clarity, I recommend replacing “physics-based simulations” to “Fluid Dynamics Simulations”. Alternatively, the authors could clarify whether they intend to generalize their approach to other physics-based simulations beyond fluid dynamics or provide examples of how the method could be applicable to other physics domains.

**Questions:**

- Could you theoretically or intuitively clarify the effectiveness of the pre-trained diffusion model in the IR sampling procedure at noise level $t = \gamma $? The distribution $x_{init}$ at noise level $t = \gamma $ appears to differ from the distribution $x_0$ at the same level $t = \gamma $. Additionally, in Equation 6, is it ensured that the error between the final output and the $x_0$ remains sufficiently small, such that $E[‖x _0^N - x_0 ‖_2 ]<\epsilon $?

- Could you also clarify (experimental support would be helpful) line 288 (2), which states that IR sampling does not require data consistency to $\hat{x}_0$? I mean why the proposed method does not require the data-consistency? Enforcing data consistency to $\hat{x}_0$ could be plugged in after line 7 in Algorithm 2 to improve accuracy in a single iteration, without compromising the number of iterations needed. Also, related to the above question, data consistency could help reduce the error towards zero, such as [1] and [2].

[1] A physics-informed diffusion model for high-fidelity flow field reconstruction, 2023.

[2] Diffusionpde: Generative pde-solving under partial observation, 2024.

**Details Of Ethics Concerns:**

See above.

---

> ### Author Response · Authors · 2024-11-21
>
> We thank the reviewer for their comments and we address their concerns below.
>
> **Superiority of the methods over other surrogates, including NOs.** We thank the reviewer for their comment on benchmarking against neural operators. As shown in Tables 6, 7, and 8, we systematically compared our methods with various deep learning baselines, including neural operators (namely FNOs) and UNets. Notably, our transient cases feature periodic boundary conditions, a scenario where FNOs are expected to excel due to their inherent architecture. In Table 6, we outline how our methods clearly outperform these baselines. Since we find from the $Tra$ case that FNOs yield suboptimal results, as corroborated by [1-3], we don't consider them for subsequent experiments.
> Further, our primary objective is to enhance the performance of DDPMs to reduce the gap between DMs and deterministic baselines; therefore, our methods don't always outperform UNets, especially when trained with advanced learning techniques. However, we demonstrated that TSM surpasses the best UNets (Table 1 left) while IR outperforms the baseline UNet (Table 1 right). In both cases, TSMs and IR surpass DDPMs in terms of speed and accuracy.
>
> [1] Benchmarking Autoregressive Conditional Diffusion Models for Turbulent Flow Simulation. In: ICML (2024).
>
> [2] Learned Simulators for Turbulence. In: ICLR (2022).
>
> [3] PDE-Refiner: Achieving Accurate Long Rollouts with Neural PDE Solvers. In: NeurIPS (2023).
>
> **Comment regarding the title.** We agree that the title may imply broader applicability than what is covered in our experiments. To address this, we will revise the title to emphasize our focus on fluid dynamics.
>
> **Effectiveness of the pre-trained DM in IR.** We thank the reviewer for raising an interesting point and *we assume that they mean $t = \gamma_{i,k}$, where $k$ denotes the $k$-th element in a refinement schedule $\gamma_i$.* The effectiveness of a pre-trained DM in IR sampling at any noise level $t$ lies in its ability to approximate the posterior mean $\mathbb{E}[\hat{x_0}| x_t]$ using Tweedie's formula, which directly relates to the likelihood estimation. In addition, we agree that in some cases the distribution $q(x_{init}|x_t)$ might not match the distribution $q(x_{0}|x_t)$ when using a pre-defined forward diffusion ($q(x_l|x_t)$ where $t > l$). Consequently, this places a limitation for the choice of $x_{init}$ to ensure that the forward process posterior estimation is optimal at any noise level $t$.
>
> **Comment regarding Eq. (6).** Eq. (6) is an idealization of a refinement schedule $\gamma$ for which each step supersedes the accuracy of the previous step and by extension all the preceding ones. This is not a hard requirement for IR sampling, but rather a consequence of the greedy optimization algorithm we employ for $\gamma$. While this doesn't guarantee that for the last refinement step the error will be smaller than a certain threshold $\epsilon$ as this depends on several factors, it is possible to provide a convergence guarantee. Our IR method is closely related to DDIM, and thus can be interpreted via ODE integrators and guaranteed to converge given an optimized refinement schedule $\gamma$. In fact, the recursive sampling formula for IR, as presented in Algorithm 2, can be recovered from the generalized generative process (see Eq. 7) that supports both Markovian and non-Markovian inference processes by using $\sigma_t = \sqrt{1-\bar{\alpha}_{t-1}}$. This essentially means that IR sampling is a special generative process similar to DDIMs (when $\sigma_t = 0$) and DDPMs, which can be recovered with a proper choice of $\sigma_t$. Thus, IR belongs to the non-Markovian family of generative processes, to which DDIM belongs.
>
> **Data consistency in IR.** We appreciate the reviewer’s suggestion regarding data consistency to improve the accuracy of IR sampling. Although it is possible to apply data consistency after line 7 in Algorithm 2, our experiments demonstrate that the proposed IR method achieves sufficient accuracy without requiring this additional step, thereby maintaining low computational cost. Enforcing data consistency with respect to $\hat{x}_0$ would require ensuring that $\hat{x}_0$ is a physically meaningful, non-noisy prediction of the flow field, which is exclusively achieved for noise steps close to 0 [4]. Additionally, to enforce data consistency, an auxiliary network or an expensive calculation of the governing PDE residual through high-order derivative approximations would be required as detailed in [5]. Thus, by not requiring data consistency in IR, the algorithm gains flexibility in selecting the noise steps for sampling (i.e., choosing the refinement schedule $\gamma$) while still allowing for future extensions that incorporate data consistency, if desired.
>
> [4] Freedom: Training-free energy-guided conditional diffusion model. 2023.
>
> [5] A physics-informed diffusion model for high-fidelity flow field reconstruction. 2023.

---

> ### Author Response · Authors · 2024-11-28
>
> Dear Reviewer,
>
> Thank you for your earlier comments and insights.
>
> We value your feedback greatly and want to ensure that our responses have addressed your concerns. To this end, we invite you to review the updated version of our paper, which includes new baselines, additional analysis, and improvements based on other reviewers' suggestions. If you have any further questions or additional feedback, we would be happy to address them.
>
> We sincerely appreciate your time and valuable feedback in helping us refine our work.

---

### Official Review · Reviewer_8zxM · 2024-10-30

**Soundness:** 3
**Presentation:** 3
**Contribution:** 3
**Rating:** 8
**Confidence:** 4

**Summary:**

This paper studies the application of diffusion models to physics simulations. Over the past years, neural networks have emerged as surrogate modeling approach for physics simulations, with a key use-case being computationally efficient inference. However, for this purpose, diffusion models have the drawback of requiring many function evaluations due to their iterative ancestral sampling procedure. To this end, the authors propose two contributions: (1) truncation of the last steps of the reverse diffusion process, and (2) iterative refinement, which considers a much shorter noise schedule at inference time. Both methods reduce the number of function evaluations and thereby increase sampling speeds. Moreover, the empirical results demonstrate that accuracy is generally maintained and sometimes even improved compared to standard expensive sampling procedures.

**Strengths:**

**S1:** The experimental setup is rigorous, comparing methods in both pointwise metrics and relevant physics-based metrics to provide complementary perspectives on their performance for three relevant datasets. Moreover, many additional results and baselines can be found in the appendix, making an extensive empirical evaluation overall.

**S2:** Two methodological contributions are evaluated: truncation of the last steps of the reverse diffusion process, and iterative refinement, which optimizes the inference sampling schedule such that less denoising steps are required. Both contributions are aimed at reducing the number of function evaluations to improve computational complexity of diffusion models for physics simulations. This is a relevant research direction, since reducing computational complexity of computational procedures is one of the primary use-cases of neural simulation models, for which diffusion is emerging as a promising modeling approach.

**S3:** The paper is well-written, and the explanations of the proposed algorithms are intuitive and easy to follow and understand. The clear structure of the text helps the reader to efficiently navigate the paper.

**Weaknesses:**

**W1:** While reading the text, I found it difficult to distill what the key differences between iterative refinement and PDE refiner are (Lippe et al., 2023). Does it have to do with the greedy optimization method of the refinement schedule (to my knowledge PDE refiner uses a fixed schedule), or details in the formulation of the diffusion process (a nonzero vs zero drift term in IR and PDE refiner respectively), or something else? Since both IR and PDE refiner are quite similar, it would be good if the ‘method novelty’ paragraph explicitly contrasts the two approaches and highlights their differences. Additionally, if the greedy optimization of $\gamma$ is a core novelty relative to PDE refiner, then it would be beneficial to explain this more elaborately in the main text rather than the appendix, since it would be a key aspect of one of the contributions in this case.

**W2:** One of the goals of the paper is to show that the proposed methods close the gap between the diffusion models and deterministic baselines. However, most of the results in the main text (both tables and plots) focus only on diffusion models. It would be relatively straightforward to also show the results of one or two deterministic methods that are considered by the authors in part of the plots and tables, for example in Figure 2 and Table 2. This would help the reader to get a better understanding of the tradeoffs of existing diffusion-based approaches, deterministic approaches, and the proposed methods without taking additional space in the paper.

**W3:** The conditioning on the autoregressive step size (j in Sec. 2 of the paper) is already introduced in Gupta et al (2022), and as such cannot be claimed as a contribution of the paper (currently point 1 of the contributions listed in the introduction). Since this is not a core point in the rest of the paper, it seems that this can straightforwardly be removed from the list of contributions in the introduction without affecting the rest of the work and the core contributions significantly.

**References:**

Gupta, J. K., & Brandstetter, J. (2022). Towards multi-spatiotemporal-scale generalized pde modeling. arXiv preprint arXiv:2209.15616.

Lippe, P., Veeling, B., Perdikaris, P., Turner, R., & Brandstetter, J. (2023). Pde-refiner: Achieving accurate long rollouts with neural pde solvers. Advances in Neural Information Processing Systems, 36.

**Questions:**

**Q1:** Given that PDE-refiner is conceptually relatively similar to iterative refinement, I am surprised to see a quite large performance difference between the two methods in Appendix D.1. Can the authors explain the reasons behind this large performance gap?

**Q2:** The truncation of the last steps of the reverse diffusion process seems to be equivalent to a modification of the noising schedule: we can choose the noise schedule $\beta_t$ such that the first step in the forward diffusion process has already a quite low signal-to-noise ratio (in line with the level corresponding to the last step before truncation in the reverse process), and afterwards noise is added gradually as per usual, while reducing the total amount of steps in the forward process in line with the skip percentage. Can the authors provide their thoughts on this perspective and whether or not they agree? If they agree, can they comment on why the noise schedule that is equivalent to the truncated process is a good choice for this problem setting relative to other problem settings, and in this way place their contribution in the broader context of noise schedules?

**Q3:** Please comment on W1-W3.

---

> ### Author Response · Authors · 2024-11-21
>
> We would like to thank the reviewer for their thorough feedback and comments regarding our work, especially for their appreciation of the experimental setup and our approaches for improved sampling.
>
> **W1 & Q1 (comparison to PDE-Refiner):** We appreciate the reviewer’s comment and agree that explicitly contrasting IR with PDE-Refiner would strengthen the clarity of our method's novelty. Since PDE-Refiner shares a lot of similarities with DDPMs in general, it is expected to share key similarities with other models/samplers including, DDIM, EDM, and IR. We summarize the main differences between IR and PDE-Refiner in the following points:
> - IR is a sampling algorithm that works with pre-trained DDPMs, while PDE-Refiner requires training from scratch for a fixed schedule and number of refinement steps, similar to DDPMs.
> - PDE-Refiner predicts the state at the initial step, while IR focuses on predicting the noise throughout the entire refinement process, similar to ancestral sampling.
> - The combination of a flexible $\gamma$ and $x_{init}$ sets IR sampling apart from PDE-Refiner as it can be treated as a standalone sampling algorithm when $x_{init} \sim \mathcal{N}(0,I)$, but can also be used to refine a noisy or a low-fidelity state.
>
> Furthermore, we think that the main reason behind the poor performance by PDE-Refiner in the $Tra$ case is that it was found to be highly sensitive to its key hyperparameters (number of refinement steps and the minimum noise variance) as reported in Kohl et al. (2024). This makes an efficient hyperparameter tuning for this method difficult, leading to suboptimal results.
>
> Moreover, while the refinement schedule $\gamma$ is one of the key novelties of IR, the greedy optimization algorithm is an established optimization approach; thus, we believe that the details regarding this algorithm are secondary to the main contributions which we include in the main text.
>
> **W2:** We thank the reviewer for their suggestion. While we do include a comprehensive list of baselines in our full set of results in Tables 6, 7, and 8, we will make sure to include the top-performing of these baselines in our Figures as well for the updated version of our paper.
>
> **W3:** We thank the reviewer for pointing out a similar approach adopted in Gupta et al. (2022) regarding our reformulated autoregressive problem. While it is true that conditioning on the prediction stride $j$ has been previously explored, the focus in the referenced study lies in evaluating models conditioned on multiple parameters, including $j$, across datasets. In contrast, our contribution emphasizes how this conditioning enables flexible sampling that supports parallelization and enables the prediction of intermediate states without compromising accuracy compared to next-step surrogates. Therefore, we will discuss the work of Gupta et al. (2022) in our revised manuscript and delineate the key differences between our objective and theirs.
>
> **Q2:** We appreciate the reviewer’s perspective on reinterpreting the TSM algorithm as a modified noise schedule with equivalent noise steps. For diffusion models in general, optimizing the noise schedule is a critical task that shall be tuned on a case-by-case basis whether through fixed schedules (e.g., linear, cosine, or sigmoid) [1] or learned ones [2]. While the proposed equivalence between truncation and a modified noise schedule might hold conceptually, in our experiments, we reported that noise schedules with large steps often produce noisy outputs, as shown in Tables 6 and 7 for the "DDPM T20" model. Therefore, adopting a custom schedule with the same noise steps as in TSMs without using Tweedie's formula for the last step at low SNR instead of an iterative refinement step is highly anticipated to result in suboptimal, noisy results. We believe that the ability to take a significant step from a low SNR state directly to a clean sample is facilitated by Tweedie's formula as it estimates the posterior mean rather than following the solution trajectory of the probability flow ODE/SDE. While this approach aligns conceptually with methods that use Tweedie's formula for final denoising (e.g., [3]), the novelty in our work lies in applying Tweedie's formula to significantly noisier states, enabling efficient sampling while maintaining accuracy.
>
>
> [1] Ting Chen. On the Importance of Noise Scheduling for Diffusion Models. 2023.
>
> [2] Kingma et al. Variational Diffusion Models. NeurIPS 2021.
>
> [3] Score-Based Generative Modeling through Stochastic Differential Equations. In: ICLR (2021).

---

> > ### Comment · Reviewer_8zxM · 2024-11-25
> >
> > Thank you for the rebuttal. Please find my response below:
> >
> > **W1 & Q1:** Thank you for the clarification on the differences between IR and PDE-refiner and updating this in the paper.
> >
> > With regard to the greedy optimization, are you aware of similar approaches being used to optimize a sampling schedule for (pre-trained) diffusion models? If this is the case, it would help to provide a citation. If this is not the case, in my opinion the paper would benefit from giving this more prominence: although greedy optimization in itself is not new (obviously), I think the way it is applied in this context is non-trivial and could inspire future work. Of course, it is up to the authors to decide.
> >
> > **W2:** Thank you. I saw that you included those results in the appendix. Given that your contribution aims to "reduce the gap between DDPMs and deterministic single step approaches", I think it would be beneficial to put some attention on the top deterministic baseline(s) in part of the figures/tables in the main text. If I am not mistaken, at this point there is only the UNet in table 1 in the updated manuscript, but I trust that the authors will show deterministic baseline results if/where space permits.
> >
> > **W3:** Thank you for the explanation and adding this to the paper.
> >
> > **Q2:** Thank you for the explanation.
> >
> > Some reviews expressed a concern that the contributions are specialized towards fluid dynamics. Although I agree, in my view, this specialization is not a problem, as the ML community focusing on this area is growing. This is evidenced by many papers accepted in major ML conferences that have had a similar focus. Further, the scope of the contribution of this work is in line with what is expected from a paper in this area.
> >
> > More importantly, this paper provides interesting and novel insights into sampling from diffusion models for fluid dynamics simulation. These insights come at the right time, as diffusion models have been gaining more traction for neural-network driven physics simulations over the last year or so. For these reasons, and with the rebuttal having alleviated my key concerns, I raised my score.

---

> > > ### Author Response · Authors · 2024-11-26
> > >
> > > We greatly appreciate the reviewer’s thoughtful feedback and their recognition of our contributions.
> > >
> > > **Greedy algorithm:** Although we think that the choice of a simple greedy algorithm is highly effective in our experiments, it may not (always) be optimal. An alternative approach for optimizing the sampling schedule of a pre-trained DM is, for example, [Bespoke solvers]( https://arxiv.org/abs/2310.19075), which may yield better results compared to our approach. Nonetheless, we agree that highlighting the effectiveness of a simple algorithm in optimizing sampling schedules is crucial in our work. We will, therefore, consider emphasizing that in our final paper.
> > >
> > > **Top deterministic baselines:** Thank you for your suggestion. We agree that it is important to highlight the performance of our models in comparison to deterministic baselines in the main text. Currently, we include UNets in our tables as they represent the top-performing baselines among those considered, making them a strong and adequate reference for comparison. As suggested, we will explore the possibility of incorporating additional baselines into our tables and/or figures, within the permitted space, to provide a clearer comparison of the various models.

---

### Official Review · Reviewer_PMEd · 2024-11-02

**Soundness:** 3
**Presentation:** 3
**Contribution:** 4
**Rating:** 8
**Confidence:** 4

**Summary:**

This article first analyzes DDPM and finds that after an appropriate truncation (stop diffusion), the model has high fidelity and high-efficiency sampling performance. On this basis, an iterative refinement method is introduced to further improve accuracy and long-term stability.

**Strengths:**

Advantages:

1. As we all know, if we consider an infinite boundary heat equation (diffusion process), or consider long-term diffusion, then the usefulness of a long part of the noise addition/noise reduction behavior is not that great. The article fully considers and utilizes the diffusion behavior in a finite time (truncation), thus achieving a balance between accuracy and running time, which is a very good point.

2. Truncated sampling reduces the uncertainty of the sample to a certain extent, or increases the accuracy of the sample response distribution.

3. Based on the content of the appendix(in particular, part D), the experimental effect is very significant. In other words, iterative refinement even makes up for the truncated sampling to a certain extent (the useful information and samples that are truncated).

**Weaknesses:**

Disadvantages:

1. I want to know whether there is a mathematical inference for the truncation sampling standard, or whether it is completely based on experience, that is, truncation and retention (importance) interpretability.

2. The purpose of refinement iteration and truncation sampling is to improve efficiency while ensuring a certain degree of accuracy. I think this requires a game. How to achieve such a balance? Is there a more rigorous mathematical explanation?

3. I think experiments can increase the breadth. One is to compare with a more general SDE instead of just with DDPM (and Kohl's 2024 work). In addition, for experiments, do you consider more general PDE solutions?

**Questions:**

The experimental results are very good, and I hope to add more theoretical analysis.  I will be happy to improve my score in subsequent discussions.

**Details Of Ethics Concerns:**

None. This is original work and there are no ethical issues

---

> ### Author Response · Authors · 2024-11-21
>
> We thank the reviewer for their comments and for highlighting the main advantages of our approaches in tackling efficient sampling with high accuracy and stability.
>
> **Mathematical interpretation of TSMs.** We appreciate the author's comment regarding the theoretical roots of TSMs. Truncation at low $t$ (i.e., using $s<0.2$) even for pre-trained DDPMs mitigates computational overhead without accuracy loss because the retained steps sufficiently capture the large scale and fine structures of the flow fields. At high $t$ (i.e., using $s\gg0.2$), an accurate approximation for the posterior mean $\mathbb{E}[\hat{x}_0|x_t;\theta]$ is only achievable by a model well-trained to approximate the score function (i.e., the gradient of the log-likelihood) for all $t \in [s\cdot T, T]$ and a target data distribution that permits a relatively simple reverse process, enabling high $s$ values. In fluid dynamics, the target distributions are often unimodal, which reduces the risk of bias introduced by Tweedie's formula, since the dominant mode of the distribution is captured with a relatively coarse discretization of the reverse-time SDE. This property supports truncation as it simplifies the sampling process without compromising the ability to accurately represent the underlying physics.
>
> **Achieving a balance between speed and accuracy in IR/TSM.** Thank you for raising this important point. Indeed, achieving a balance between speed and accuracy in IR sampling and TSMs requires careful tuning of the associated hyperparameters. These are dataset- and task-dependent parameters that need to be optimized on a case-by-case basis to serve the specific dynamics of the problem.
> Regarding the efficient optimization of these hyperparameters, we provide a heuristic approach supported by our results:
>
> - Deterministic test cases
>     - IR: we start with $x_{init} \sim \mathcal{N}(0,I)$ and run our greedy optimization algorithm with low N (with $N = |\gamma|$) to obtain an efficient $\gamma$ with low NFEs. As in Figure 4(b), $N=5$ often yield very good results for transient cases. $N$ can then be gradually increased to explore other schedules that could potentially enhance the accuracy with mild increase in NFEs.
>     - TSM: For higher speedup, the search for the optimum $s$ value typically begins within the range $[0.5, 1]$, especially if a large number of diffusion steps $T$ is chosen. We believe that Fig (3) provides several insights for the optimal combination of $s$ and $T$. We first test for both extremes of $s$ and then follow a standard line search approach to arrive at an optimum value for $s$, requiring additional 2 or 3 evaluations at most. We also give priority for models with low $T$ to minimize the NFEs required for inference.
> - Stochastic test cases
>     - IR: $x_{init}$ is optimal when obtained through truncated sampling of a pre-trained DDPM with $s >= 0.5$. The output from this approach is usually highly accurate but exhibits clear noise; thus it usually takes $N < 5$ for $\gamma$ to arrive at a clear output while improving or retaining the accuracy of the noisy input.
>     - TSM: We follow the same procedure as before; however, we refrain from extreme $s$ values. Therefore, our search is restricted to $s \in [0.5, 0.9]$, or smaller lower bound for low $T$ values.
>
> While a rigorous closed-form mathematical framework remains challenging, our heuristic approach was found to provide adequate results for our experiments, minimizing typical exhaustive parameter search. We will expand on this in the revised paper to improve clarity.
>
> **Increasing breadth of experiments.** We appreciate the reviewer’s suggestion to broaden our experimental scope. We have included an additional comparison against EDMs for 3 of our experiments. In summary, across different fluid dynamics problems, EDMs demonstrate varying performance. On the $Tra$ dataset, EDM achieves the best accuracy with only 4 NFEs, surpassing all probabilistic and deterministic models. TSM achieves nearly comparable accuracy while remaining the sole model capable of single-step inference. On $Fturb$, EDM outperforms DDPM and DDIM in both accuracy and NFEs but achieves comparable performance to TSM, despite being $10\times$ slower, whereas IR remains the most accurate model for this dataset. However, on $Air_{One}$, EDM underperforms compared to all other models, likely due to suboptimal hyperparameter settings, including loss weighting, scaling, and diffusion parameters, which may require extensive tuning. Please see our response to reviewer *6cX6* for additional details on the EDM comparisons.
>
> Regarding additional PDE solutions, while we believe that the current experiments cover diverse and intricate fluid dynamics phenomena, including both deterministic and stochastic scenarios, we acknowledge the value of testing on broader PDE settings. We’d be happy to include an additional experiment on the 1D Kuramoto-Sivashinsky problem in the camera-ready version of our paper.

---

> > ### Comment · Reviewer_PMEd · 2024-12-01
> >
> > Your answer has cleared my doubts to a certain extent. It seems that the Truncation Model does have broader advantages. I am very grateful and I have improved my score.

---

> ### Author Response · Authors · 2024-11-28
>
> Dear Reviewer,
>
> Thank you for your earlier comments and insights.
>
> We value your feedback greatly and want to ensure that our responses have addressed your concerns. To this end, we invite you to review the updated version of our paper, which includes new baselines, additional analysis, and improvements based on other reviewers' suggestions. If you have any further questions or additional feedback, we would be happy to address them.
>
> We sincerely appreciate your time and valuable feedback in helping us refine our work.

---

### Official Review · Reviewer_TSeh · 2024-11-03

**Soundness:** 3
**Presentation:** 3
**Contribution:** 2
**Rating:** 6
**Confidence:** 4

**Summary:**

The paper introduces a truncation approach and an iterative refinement process in the sampling procedure of the Denoising Diffusion Probabilistic Model (DDPM) that enable to reduce the number of function evaluations without decreasing the accuracy. The first method proposes to stop the sampling process at an earlier time point and to estimate the denoised sample using Tweedie's formula. The second method uses the forward diffusion for a given shorter noise schedule and the denoised sample is approximated using Tweedie's formula. The authors show the efficiency of the approach on the simulations of airflow field.

**Strengths:**

The paper reads very well and provides some good contributions to the development of diffusion models.

   - Although the presented approach is based on some known results (Tweetie's formula, ancestral sampling), the proposed solution seems very efficient in practice and leads to lower computational costs. Both idea's exploit the denoising Tweedie's formula in forward and backward sampling.

   - Including the truncation in the training is smart trick that eases the training.

   - The numerical experiments show that the presented approaches achieve better performances than traditional DDPM sampling and reduce the computational costs.

- The approach seem to be very efficient in physic based simulation of flow field. I think this is a very good contribution in this specific domain.

**Weaknesses:**

Although the approach seems efficient in the presented numerical experiments, there is a number of points that need to be clarified.

- Tweedie's formula is well known. It seems that this has been already used in some previous works such of [Delbracio et Milanfar, 2024]. In  their work, the authors provide some intermediate reconstructions through this formula. They show that by adding some stochastic steps
they can get better performances than state-of-the-art. I know the model is not the same but it would have been interesting to highlight the links with this work because they seem very closely related. May the authors comment on that point?

- The results show that TS outperforms traditional DDPM by using s=1. This means that on this specific problem, there is no need to sample intermediate diffusion steps. I wonder whether this aspect is problem specific or this happens for a wider range of problems. Is this result expected?

- This would have been interesting to see how this method perform against traditional DDPM sampling on image reconstruction. Indeed, DDPMs usually perform well on such problems.

- The training data are deterministic sequences of flow field data. It would have been interesting to observe how the model performs on noisy dataset.

- Overall, I like the idea of truncation and iterative refinement, but since there is no major theoretical contribution in this work, I would have liked to see more numerical results. The claim "Truncation is all you need" would have been justified if the authors had included numerical results on different applications. So far, the paper makes very good contribution in this specific domain, and proposes an interesting approach to reduce computational costs.

Delbracio et Milanfar, 2024, Inversion by Direct Iteration: An Alternative to Denoising Diffusion for Image Restoration. TMLR.

**Questions:**

- Does s=1 mean that the model is trained as a single variational autoencoder? Or is this truncation only used for sampling? (paragraph about the training is not clear to me)

- naive question for my understanding: At the beginning of Section 5.1, the authors say that they consider time series of flow field from j= 1 to T. It is not clear how the time series are handled here. Could the authors clarify this point?

- How do the authors handle the high fluctuations areas of the domain? It seems that some region of the domain have a highly turbulence flow field (low pressure vortex) and this would require a more flexible model in this specific area.

- Did the authors try to change the value of the initial input $x_{init}$?

I look forward to reading the answers of the authors and I can change my score depending on their answers.

---

> ### Author Response · Authors · 2024-11-21
> **Reply (1/2)**
>
> We thank the reviewer for their thoughtful comments and their interest in our approaches to reduce computational costs. We address their concerns below.
>
> **Use of Tweedie's formula and similarity to [Delbracio et Milanfar, 2024].** We thank the reviewer for highlighting the connection between our proposed approaches, leveraging Tweedie's formula, with [Delbracio et Milanfar, 2024]. As mentioned, the models are fundamentally different, yet the aforementioned work can still be linked to DDPMs when the low-quality, degraded state is pure Gaussian noise. Their main reason to add scaled white noise in every step of their inference algorithm is to convert a deterministic algorithm to a stochastic one that is capable of exploring multiple possible explanations for a degraded sample and thus lead to better perceptual quality, though their approach was shown to not be beneficial in all cases. Since both our approaches are inherently stochastic (i.e., fresh white noise is injected in every step), the potential gains from injecting further white noise to the predicted posterior mean might have little effect. Alternatively, we can enforce data consistency after applying Tweedie's formula to improve the overall accuracy of the methods similar to resampling methods (e.g., [1] and [2]). However, this approach comes with its own limitations as we described in the paragraph starting at line 284.
>
> [1] Solving inverse problems with latent diffusion models via hard data consistency. In ICLR (2024).
>
> [2] Improving Diffusion Inverse Problem Solving with Decoupled Noise Annealing. In: CoRR (2024).
>
> **Superior performance of TSM with $s=1$.** We would like to clarify that single-step TSMs (i.e., using $s=1$) don't always outperform ancestral sampling. $s$ is a hyperparameter that is to be tuned on a case-by-case basis and it is not always guaranteed that extreme $s$ values will always lead to the best results as they can significantly impact the stochasticity of the method. Since complex datasets generally benefit from stochastic samplers [3], we see that $s = 1$ is only possible on the $Tra$ (see Table 6) case while the optimum results for the cases $Fturb$ and $Air$ are obtained using $s = 0.75$ (see Table 7) and $s = 0.6$ (see Table 8), respectively. This is due to the increasing level of difficulty in these test cases, requiring more steps of the reverse Markov chain. Therefore, the trade-off between accuracy and stochasticity would limit the practicality of perpetually using $s = 1$. The observed results are specific to the nature of fluid dynamics simulations and thus, a generalization is only possible when considering the features of the data distribution (as we explain in the paragraph beginning at line 406).
>
> [3] Karras et al. “Elucidating the Design Space of Diffusion-Based Generative Models”. In: NeurIPS (2022).
>
> **TSM performance in image reconstruction.** We appreciate the reviewer’s interest in benchmarking our approaches against traditional DDPM sampling on image reconstruction. However, our focus on fluid dynamics problems is intentional due to the unique properties of the data. Physics-based simulations involve deterministic and continuous systems governed by PDEs, which differ significantly from the more complex, often multimodal distributions encountered in generative modeling tasks. Our methods are tailored to exploit these characteristics present in our datasets to support fast and accurate sampling of DMs. Hence, comparing TSMs/IR against ancestral sampling for image reconstruction is beyond the scope of our research.
>
> **Models performance on noisy dataset.** Our primary objective from this study is to evaluate the effectiveness of our proposed methods in capturing the underlying dynamics of fluid systems without introducing additional complexities. We believe that handling noisy datasets is an independent, challenging task, potentially requiring methodological modifications to the training or the denoising algorithms. For example, existing studies are dedicated to exploring analogous challenges, including noisy [4] or sparse [5] observations. One simple approach is to include the conditional information (i.e., the previous noisy state) in the iterative refinement process similar to [6], but the efficacy of this method in reducing the impact of added noise and its ability to recover a noise-free observation remains uncertain. Thus, we believe that out-of-the-box diffusion models are not expected to excel on noisy datasets without careful algorithmic modifications, which is an interesting topic for future work.
>
> [4] Risk-Sensitive Diffusion: Robustly Optimizing Diffusion Models with Noisy Samples. 2024.
>
> [5] DiffusionPDE: Generative PDE-Solving Under Partial Observation. 2024.
>
> [6] Kohl et al. Benchmarking Autoregressive Conditional Diffusion Models for Turbulent Flow Simulation. In: ICML (2024).

---

> > ### Author Response · Authors · 2024-11-21
> > **Reply (2/2)**
> >
> > **Comment regarding the title.** We appreciate the reviewer’s positive feedback on our truncation and IR approach. We acknowledge that our title, “Truncation is all you need,” could be misleading to the non-specialist audience as it implies broader generalizability across domains beyond fluid dynamics. As suggested, we will revise it to more accurately reflect our focus on physics-based simulation tasks.
> >
> > > Q: Does s=1 mean that the model is trained as a single variational autoencoder? Or is this truncation only used for sampling? (paragraph about the training is not clear to me)
> >
> > Indeed, $s$ is used both during training and sampling; thus, when $s=1$, this corresponds to single-step inference. The noise step $t$ condition for the network in this case can be omitted during training and sampling.
> >
> > > At the beginning of Section 5.1, the authors say that they consider time series of flow field from j= 1 to T. It is not clear how the time series are handled here. Could the authors clarify this point?
> >
> > We would like to clarify that $j$ is the prediction stride that defines the temporal intervals at which we sample future states of the flow field based on the physical timestep $\delta \tau$. It is part of our reformulated autoregressive sampling method detailed in Section 2.2. It is a discrete parameter that takes values between 0 and $\mathcal{T}$ (i.e., $\mathcal{T} + 1$ possible values) that is used as a condition for the network and is independent from the fluid flow sequences with $R$ timesteps. To reach the target time $\tau_f$ which is $R$ steps from the initial condition, we can take up to $\lceil R/j \rceil$ steps, which is maximum when $j=1$ (i.e., next-step prediction) and minimum when $j=\mathcal{T}$ (i.e., temporal stride of $\mathcal{T}$ steps).
> >
> > > How do the authors handle the high fluctuations areas of the domain?
> >
> > We appreciate the reviewer’s concern regarding high-fluctuation regions which are definitely critical in fluid dynamics modeling. Typically in DDPMs and other deep learning-based surrogates, and by extension our approaches, we don't explicitly target specific physical phenomena; instead, we rely on the learning process to capture these dynamics intrinsically. By training our models on high-fidelity datasets, we ensure they learn the data distribution and the underlying physical characteristics of the flow field, including turbulent structures such as vortices.
> >
> > > Did the authors try to change the value of the initial input $x_{init}$?
> >
> > For the $Tra$ and $Fturb$ test cases, we found that IR with $x_{init} = x_T$ provides highly accurate predictions with low NFEs; therefore, we didn't consider other $x_{init}$ values that might incur extra computational cost without adequate increase in accuracy. In the $Air$ cases, we experimented with $x_{init} = x_T$ as well as values obtained with truncated ancestral sampling (i.e., a pre-trained traditional DDPM sampled using Algorithm 1). We found the latter to achieve the most accurate predictions with the least possible NFEs.

---

> > > ### Comment · Reviewer_TSeh · 2024-11-25
> > > **Reply to Authors**
> > >
> > > Thank you for the clarifications and the detailed answers.
> > > As a general comment, I find the answers convincing about the quality of the paper. (I thank the authors for taking into account the recommendations for the title of the paper. This title reflects the content of the paper.)
> > > However, I still think that paper's contributions are limited to specific flow field simulation improvements and I agree with Reviewer 6cX6 about the lack of theoretical guarantees. Providing theoretical convergence guarantees on both algorithms would have made the paper more impactful. For the moment, this paper could be a very nice a paper in any fluid dynamic journal but the machine learning contributions are slightly low for a machine learning conference. In my opinion, the authors should investigate the convergence properties (we might be able to derive some bounds from Tweedie's formula)  and the potential applications to some other datasets. It is not far from a good paper for ICLR and the questions raised by this paper are very interesting.

---

> > > > ### Author Response · Authors · 2024-11-25
> > > >
> > > > We greatly appreciate the reviewer’s thoughtful feedback and acknowledgment of the quality and relevance of our work. Our focus on fluid dynamics is intentional since fluid dynamics problems are prevalent and because we identify a gap where DMs have underperformed in this domain, primarily with respect to inference speed, as we outline in the paragraph starting at line 54. Therefore, our methods target improvements in this particular domain to reduce this gap by utilizing the unique data distribution characteristics of these simulations. We thus believe that our methods can generalize to other physics-based simulations beyond fluid dynamics if they exhibit similar distributions features. This can indeed be tackled in future studies. Moreover, we would like to emphasize that our contributions address a pressing need in temporal generative modeling and constitute meaningful progress for the machine learning and fluid dynamics communities alike.
> > > >
> > > > Regarding theoretical guarantees, we would be happy to investigate if error bounds can be derived from Tweedie's formula as suggested and share any useful outcomes in our final paper.

---

### Author Response · Authors · 2024-11-21
**Global Reponse**

We greatly appreciate the reviewers' positive comments on our methods' significant contribution to efficient sampling in fluid dynamics, as supported by extensive experiments, as well as the constructive feedback provided by all reviewers, which has been invaluable in refining our work. We acknowledge the concerns and suggestions raised, and we have made several revisions to address these points, enhancing the clarity and breadth of our contributions.

We summarize the major upcoming updates which will be included in our revised manuscript in the next couple of days:

- We shall update the title to emphasize our focus on fluid dynamics instead of general physics-based simulations. We propose the title "Improved Sampling of Diffusion Models in Fluid Dynamics Simulations by Leveraging Tweedie's Formula," as it more precisely conveys the focus of our study.
- We will include our newly evaluated EDMs for 3 out of 4 experiments, comparing their performance against standard DDPMs and our proposed approaches. In summary:
    - EDMs excel on $Tra$, achieving top accuracy with 4 NFEs, while TSM follows with almost same accuracy as the only probabilistic model enabling single-step inference with high accuracy. With a more accurately pre-trained DDPM, IR can potentially achieve more competitive results given the insignificant accuracy difference from the best-performing models.
    - On $Fturb$, EDM is on par with TSM, albeit being $10\times$ slower, whereas IR remains the most accurate.
    - On $Air_{One}$, EDM clearly performs poorly, likely due to suboptimal hyperparameters.
    - We conclude that, although EDMs may outperform standard DM baselines, they are less accurate and/or slower than our proposed approaches as demonstrated across a range of fluid dynamics problems.
- As requested, we will add the results for our temporal stability analysis study for the transient cases using very long rollout trajectories.
- We will add an appendix summarizing the main parameters for all our experiments, including benchmark ones, to ensure the clarity of our experimental setup.
- As requested by reviewer *fSso*, we will extend the discussion on the novelty of IR method to highlight the key differences with PDE-refiner and why the latter performs suboptimally compared to our methods.
- An additional appendix will be dedicated to describing a heuristic approach to efficiently optimize the hyperparameters associated to our approaches supported by the findings from our experiments.

---

### Comment · Area_Chair_sG32 · 2024-11-25
**Reviewers' Response**

Dear Reviewers,

As the author-reviewer discussion period is approaching its end, I would strongly encourage you to read the authors' responses and acknowledge them, while also checking if your questions/concerns have been appropriately addressed.

This is a crucial step, as it ensures that both reviewers and authors are on the same page, and it also helps us to put your recommendation in perspective.

Thank you again for your time and expertise.

Best,

AC

---

### Author Response · Authors · 2024-11-25
**Updated Manuscript**

Dear reviewers,

We have uploaded a revised manuscript with significant modifications to include the new results and your comments. These modifications are:

- Updated the title to be “Improved Sampling Of Diffusion Models In Fluid Dynamics With Tweedie's Formula” to clearly convey the focus of the paper.
- Highlighted text modifications in red to include discussions with the reviewers.
- Updated all respective tables and figures to include the results of EDM for the three experiments.
- Added plot (Fig. 7) for EDM sampling using various sampler configurations for all considered cases.
- Added temporal stability plot (Fig. 9) for the transient problems.
- Added Appendix A (experimental setup) and Appendix G (hyperparameters tuning).

Upon acceptance, we will add the following for the camera-ready version:

- An additional experiment to evaluate our methods on the 1D Kuramoto-Sivashinsky problem as suggested by the reviewers.
- Training and evaluation of EDMs for the $Air_{Multi}$ case for comparison against our methods.
- Updating the code to include EDM training and sampling algorithms.

We thank all reviewers for their valuable feedback, and we would like to invite them to check the revised manuscript.

---

### Meta-Review · Area_Chair_sG32 · 2024-12-18

**Metareview:**

This paper introduces two techniques, truncation and iterative refinement, aimed at improving the efficiency of diffusion models by reducing the number of function evaluations while maintaining, and even increasing, the statistical accuracy of the samples. Both techniques leverage the well-known Tweedie's formula and are evaluated in steady and time-dependent CFD problems.

Reviewers highlight the paper's success in demonstrating that these techniques significantly reduce NFEs, leading to faster sampling and improved stability. The experimental validation on turbulent and airfoil flow datasets is considered appropriate and supports the claim that these methods maintain (or enhance) accuracy compared to standard DDPMs.

However, as only examples for CFD are presented, it is unclear how well these techniques would be applicable to broader ML-related tasks. In addition, the claim regarding long rollouts appears to be based on relatively short rollouts particularly when compared to recent works in this area (see [1, 2, 3]).

After considering the strengths and weaknesses of the paper, I recommend acceptance.

References:

[1] Jiang, Ruoxi, et al. "Training neural operators to preserve invariant measures of chaotic attractors." Advances in Neural Information Processing Systems 36 (2024).

[2] Schiff, Yair, et al. "DySLIM: Dynamics Stable Learning by Invariant Measure for Chaotic Systems." Forty-first International Conference on Machine Learning (2024).

[3] Li, Zongyi, et al. "Learning chaotic dynamics in dissipative systems." Advances in Neural Information Processing Systems 35 (2022): 16768-16781.

**Additional Comments On Reviewer Discussion:**

The reviewers liked the simplicity of the ideas, but they complained about the title, which was then modified. They also raised issues bout the limited scope of the paper (flow field simulation) and lack of theoretical guarantees.

---

### Decision · Program_Chairs · 2025-01-22

Accept (Poster)